# Momentum Stiefel Optimizer, with Applications to Suitably-Orthogonal Attention, and Optimal Transport

**Lingkai Kong, Yuqing Wang, Molei Tao**
Georgia Institute of Technology     `{lkong75,ywang3398,mtao}@gatech.edu`

## Abstract

The problem of optimization on Stiefel manifold, i.e., minimizing functions of (not necessarily square) matrices that satisfy orthogonality constraints, has been extensively studied. Yet, a new approach is proposed based on, for the first time, an interplay between thoughtfully designed continuous and discrete dynamics. It leads to a gradient-based optimizer with intrinsically added momentum. This method exactly preserves the manifold structure but does not require additional operation to keep momentum in the changing (co)tangent space, and thus has low computational cost and pleasant accuracy. Its generalization to adaptive learning rates is also demonstrated. Notable performances are observed in practical tasks. For instance, we found that placing orthogonal constraints on attention heads of trained-from-scratch Vision Transformer (Dosovitskiy et al., 2020) could markedly improve its performance, when our optimizer is used, and it is better that each head is made orthogonal within itself but not necessarily to other heads. This optimizer also makes the useful notion of Projection Robust Wasserstein Distance (Paty and Cuturi, 2019; Lin et al., 2020) for high-dim. optimal transport even more effective.

Code: `https://github.com/konglk1203/VariationalStiefelOptimizer`

## 1 Introduction

Matrices that satisfy orthogonal constraints play important roles in various areas including machine learning. A range of studies showed this both theoretically (Saxe et al., 2013; Xiao et al., 2018) and experimentally – for example, orthogonality can boost the performances of many architectures[1], e.g., MLP (Cisse et al., 2017), CNN and ResNet (Li et al., 2020; Wang et al., 2020), RNN (Arjovsky et al., 2016; Wisdom et al., 2016; Helfrich et al., 2018), Transformer (Zhang et al., 2021). The training of such models amounts to optimization under orthogonal constraints, whose applications to machine learning, however, are not restricted to just improving deep learning models. For example, such optimization helps construct optimal low-dim. projection of high-dim. data, which can be used for, e.g., robust and efficient approximation of Wasserstein distance (Paty and Cuturi, 2019; Lin et al., 2020). Therefore, this article will focus on this non-Euclidean / Riemannian optimization problem.

In fact, matrices that contain orthogonal columns constitute a Riemannian manifold known as the Stiefel manifold. Given integers $n \geq m > 0$, it is defined as $\mathsf{St}(n, m) := \{X \in \mathbb{R}^{n \times m} : X^\top X = I_{m \times m}\}$ (see Apdx. B.3 for the $n = m$ special case which is almost $\mathsf{SO}(n)$). Then, given a $\mathcal{C}^1$ function $f : \mathsf{St}(n, m) \to \mathbb{R}$, we consider, under gradient oracle, the smooth optimization problem

$$\min_{X \in \mathsf{St}(n,m)} f(X), \qquad \text{which is equivalent to} \qquad \min_{X \in \mathbb{R}^{n \times m}, X^\top X = I_{m \times m}} f(X). \qquad (1)$$

Due to the importance of this problem, a wide range of approaches have been proposed. Methods based on regularizers, which approximate the constraints (and hence the manifold structure), for example, have been popular in the machine learning literature (Cisse et al., 2017; Bansal et al., 2018; Wang et al., 2020; Zhang et al., 2021). Meanwhile, efforts have been continuously made to construct algorithms that truly preserve the constraints (i.e. the manifold structure), although challenges such as computational scalability and how to add momentum still remain; see the 2nd next paragraph for how our method addresses them, and more discussions on existing milestones in Sec. 1.1.

Our strategy toward a good Stiefel optimizer is the following: first, formulate a variational principle and use that to construct an optimizing dynamics in continuous time, which is described by a system

---

[1]It helped computer vision applications prior to the deep learning era as well (e.g., Liu et al., 2003).

of ODEs corresponding to damped mechanical systems on a constrained manifold; then, design a delicate time discretization of these ODEs, which yields optimization algorithms that precisely preserve the constraints and mimic the continuous dynamics.

Our optimizer has several pleasant properties: 1) It is exactly preserving the manifold structure, not only of the Stiefel manifold, but in fact of its tangent bundle. In other words, throughout the course of optimization, the position variable remains exactly on the Stiefel manifold, and the momentum variable remains exactly in the (co)tangent space. 2) Typically, in order to maintain the manifold structure, some kind of projection/retraction/exponential-map operation is needed, and since we have both position and momentum, such operation is needed for both variables (i.e. to maintain the cotangent bundle structure). However, our carefully designed ODE and its discretization make the structure preservation of momentum automatic, meaning that no extra operation (projection, retraction, parallel transport, etc.) is needed for the momentum variable. This not only leads to improved computational efficiency, but also serves an indirect evidence of having a reduced overall (i.e. both position and momentum) local error. 3) We used a quadratic-convergent iterative solver for our specific position retraction operation, which makes it fast. 4) Due to 2)+3), our per iteration computational complexity, $\mathcal{O}(nm^2)$, has a small constant factor (see Sec. C for details). 5) Our discretization is also numerically stable so that it well preserves the structure even under low machine precision and numerous iterations, which are beneficial in machine learning contexts. 6) Because our algorithm is derived from a variational framework that unify both Euclidean and Stiefel variables, the same hyperparameters can be used for both these parameters; see Sec.3 and note this difference from previous milestones (e.g., Li et al. (2020)) significantly reduces tuning efforts. 7) Our algorithm works for a range of Riemannian metrics, allowing extra flexibility in choosing suitable geometry to optimize the performance for a specific problem.

Selected (due to space) experimental tests of our optimizer are: (1) We consider the simple problem of leading eigenvalues, which is yet practically important in data sciences. It systematically investigates algorithmic performances under different parameters. (2) We show the elegant idea of approximating optimal transport distance in high-dim. via a good low-dim. projection (Paty and Cuturi, 2019; Lin et al., 2020) can be made even more efficacious by our optimizer. (3) We note that Vision Transformer (ViT) can be further improved by imposing attention heads to be orthogonal; more precisely:

Consider training ViT from scratch. We discover that 1) requiring each head to be orthogonal in itself improves both training and testing accuracies the most. An important recent work by Zhang et al. (2021) applied orthogonality to transformers and demonstrated improved performance in NLP tasks. It concatenates each of the $W_i^Q, W_i^K, W_i^V$ matrices in attention across all heads, and applies orthogonal constraint to each of the three, via regularizer. This makes each head (approximately) orthogonal, not only within itself, but also to other heads. Orthogonal constraint is also applied, via regularizer, to each weight matrix of feed-forward layers in their case. With our Stiefel optimizer which are not restricted to square matrices, we can now make each head exactly and only orthogonal within itself, which leads to further improvements at least in CV tasks. Meanwhile, 2) having orthogonality both in and across heads is found less effective than 1), but it is still better than requiring no orthogonality (i.e. vanilla ViT). No orthogonality on feed-forward layers was used in either 1) or 2). In addition, 3) to achieve these improvements, our Stiefel optimizer needs to be used; methods that do not have momentum or not exactly preserve structure (e.g., regularizer-based) are seen not fully exploiting the benefit of orthogonality.

## 1.1 MORE ON RELATED WORK

**Orthogonality in deep learning** Initialization: Orthogonal initialization is both theoretically and experimentally shown good for deep neural networks (Saxe et al., 2013). CNN: Cisse et al. (2017) used regularizer to make weight matrices orthogonal when training MLP and CNN, and showed both improved accuracy and robustness. Rodríguez et al. (2017) showed that posing orthogonality to CNN reduces overfitting. Bansal et al. (2018) experimentally compared several orthogonal regularizers on ResNet. RNN: Arjovsky et al. (2016); Wisdom et al. (2016); Helfrich et al. (2018); Vorontsov et al. (2017); Lezcano-Casado and Martınez-Rubio (2019) showed orthogonalilty helps plain RNN achieve long term memory and even out-perform advanced models such as LSTM. Transformer: Despite of its success, transformer is actually a recent model; to the best of our knowledge, the community just started applying orthogonal constraint to Transformer for NLP tasks (Zhang et al., 2021). Our work is the first to apply it to Vision Transformer (Dosovitskiy et al., 2020).

**Optimization algorithms** Manifold optimization is a profound field. General approaches for 1st-order (i.e. gradient based) optimization on (Riemannian) manifold typically involve retraction

(see Apdx.D) which is not present in Euclidean cases. Thus we categorize 1st-order Stiefel optimizers into 2 types: O1) retraction-based, and O2) non-retraction-based.

O1) Retraction-based. Given a point and a tangent vector, exponential map gives how the point moves along the tangent vector on the manifold. In practice, an approximation of the exponential map, known as a retraction, is often used for reducing the computational cost. The exponential map and various retractions on Stiefel manifold are well-studied. For Stiefel, the exponential map can be computed with complexity $\mathcal{O}(nm^2)$ (Edelman et al., 1998), whose constant prefactor is however large as the map is essentially matrix exponentiation. A well-known retraction is Cayley map (Eq. (32)), and Wen and Yin (2013) proposed a gradient descent method[2] with smartly lowered computational cost of Cayley map (will be referred to as `Momentumless Stiefel (S)GD`). SVD can also help construct a retraction; Li et al. (2019) did so by first following the gradient out of the manifold and then projecting back via modifying the singular values, which is interesting but expensive. Meanwhile, a remaining challenge is to add momentum, which is nontrivial in a manifold setting but helpful for improved speed of convergence (Zhang and Sra, 2018; Ahn and Sra, 2020; Alimisis et al., 2021). In this case, the geometry that needs to be maintained is that of the augmented state-space, known as the tangent bundle; i.e., position has to be in Stiefel manifold and momentum in its tangent space. One could follow a general Riemannian retraction idea (e.g., Absil and Malick (2012)) and use retraction that projects both position and momentum to the tangent bundle to ensure manifold preservation. This was done for example in Li et al. (2020) (will be referred to as `Projected Stiefel SGD/Adam`). Another cool technique to maintain the structure of momentum is parallel transport (e.g., Bécigneul and Ganea (2019)), which is however computationally expensive and possibly inexact (Edelman et al., 1998). Our work could be viewed as a retraction-based approach, because our position variable is projected back to St at each step; however, it doesn't require any retraction or projection on the momentum variable — thanks to our variational approach and delicate discretization, position and momentum intrinsically move together, and once the position $X$ is placed on St, the momentum $Q$ automatically finds itself in $T_X$St. This leads to better computational efficiency, as well as improved accuracy because variables stay closer to the tangent bundle.

O2) Non-retraction-based. In some sense one can always call an exact manifold preserving method retraction-based, but approximate structure preservation is also a possibility. Regularizers for instance could be used to approach $X^\top X = I$. Between exact and approximate structure preserving, which one works better is theoretically still an open question, and task-dependent but insightful empirical results exist. For example, regularizer-based methods usually need more hyperparameter tuning, and the final result is likely sensitive to the regularization strength (Wang et al., 2020). Lai and Osher (2014) also discussed their slow convergence and possible improvements. Regularizer-based methods typically have low computational costs, but due to trajectory not exactly on the manifold, they may converge to (the neighborhood of) a local min. different from the one an exact algorithm converges to for multimodal problems. Similar observation was made for non-regularizer-based approximate method too; e.g., Vorontsov et al. (2017) suggested that at least in certain cases 'hard' constraint is better than 'soft'. The aforementioned Li et al. (2020) is actually inexact too, because in their practical implementation the retraction was inexactly performed for the sake of speed. Meanwhile, we also note an interesting, non-regularizer-based recent result (Ablin and Peyré, 2022) that converges to the manifold despite of being approximate prior to convergence.

Additional. We also mention the existence of useful algorithms for electronic structure calculations that can be viewed as Stiefel optimizers (e.g., Zhang et al. (2014); Dai et al. (2017); Gao et al. (2018; 2019); Hu et al. (2019); Dai et al. (2020)). They did not use momentum. Another inexact-manifold-preserving but interesting result is Bu and Chang (2022). The problem can also be approached via a constrained optimization formulation (e.g., (Leimkuhler et al., 2021)). In addition, there are other interesting optimizers also based on discretizing damped mechanical systems but not via the splitting approach used here, e.g., Lee et al. (2021); Duruisseaux and Leok (2021); however they are mostly implicit and not always scalable to deep learning applications. Finally, it is possible to represent a matrix by other matrices and optimize those matrices instead. A special case of $\mathsf{SO}(n)$ was considered by Lezcano-Casado and Martınez-Rubio (2019); Helfrich et al. (2018) where $X = \mathrm{expm}(A)$ or $X = \mathrm{Cayley}(A)$ was used and then one optimized skew symmetric $A$ instead. Although these authors did not consider a Stiefel generalization, this could be possible, but even without the generalization the computational cost is already high.

---

[2]Their setting was deterministic, but an extension to use stochastic gradient descents is straightforward.

This paper considers exact structure preservation. For applications included here, this is either empirically more beneficial (Sec.3.2) or a necessity (Sec.3.1 & Apdx.Q). See comparisons of most relevant approaches in Tab. 2 and more on complexity in Apdx. C.

## 2    DERIVATION OF THE OPTIMIZERS AND THEIR PROPERTIES

We represent Stiefel manifold as $\mathsf{St}(n, m) := \{X \in \mathbb{R}^{n \times m} : X^\top X = I_{m \times m}\}$ where $n \geq m$[3], i.e. matrices with orthonormal columns. This is a natural embedding in Euclidean space. Based on identity isomorphism between cotangent & tangent spaces (see Apdx. B.2), it also gives:

**Proposition 1.** The *(co)tangent space* of $\mathsf{St}$ at $X \in \mathsf{St}$ is $T_X\mathsf{St} := \{\Delta \in \mathbb{R}^{n \times m} : X^\top \Delta + \Delta^\top X = 0\}$ ; its *(co)tangent bundle* is $T\mathsf{St} := \{(X, \Delta) : X \in \mathsf{St}, \Delta \in T_X\mathsf{St}\}$.

Throughout this paper, we focus on a family of Riemannian metrics on $\mathsf{St}$ defined in the following.

**Definition 1** (canonical-type metric). For a fixed constant $a < 1$, the *canonical-type metric* $g_X : T_X\mathsf{St} \times T_X\mathsf{St} \to \mathbb{R}$ is defined as

$$g_X(\Delta_1, \Delta_2) = \mathrm{Tr}(\Delta_1^\top (I - aXX^\top)\Delta_2), \quad \forall \Delta_1, \Delta_2 \in T_X\mathsf{St}. \qquad (2)$$

The following are two commonly used examples of the canonical-type metric (Tagare, 2011):

- When $a = 0$, $g_X(\Delta_1, \Delta_2) = \mathrm{Tr}(\Delta_1^\top \Delta_2)$, which corresponds to the Euclidean metric.
- When $a = 1/2$, $g_X(\Delta_1, \Delta_2) = \mathrm{Tr}(\Delta_1^\top (I - \frac{1}{2}XX^\top)\Delta_2)$, which is known as the canonical metric.

**Notation.** Denote Euclidean gradient in ambient space by $\frac{\partial f}{\partial X} := \left(\frac{\partial f}{\partial X_{ij}}(X)\right)_{ij}$. Denote by $\odot, \oslash$, and $A^{\circ c}$ element-wise product, division, and $c$th power. Denote expm to be the matrix exponential.

### 2.1    OPTIMIZATION DYNAMICS IN CONTINUOUS TIME

In this section, we study the Stiefel optimization problem (1) from a variational perspective. The manifold setup yields a constrained variational problem, which is hard to handle due to nonlinear geometry of the function space. Thus we introduce an alternative variational formulation based on functional Lagrange multiplier to bypass this difficulty. It generates an ODE that never escapes from the manifold $T\mathsf{St}$ and is guaranteed to converge to a local minimizer of the function $f$ on $\mathsf{St}$.

In Euclidean space, momentum has long been used to accelerate gradient descent (Nesterov, 1983). More recently, continuous time limit brought useful tools to complement the classical analyses for such optimizers (e.g., Su et al. (2014)), and a variational formulation was then established to unify a large family of momentum-based Euclidean optimizers (Wibisono et al., 2016). This formulation can also provide a powerful and intrinsic way to generalize momentum-based optimization to non-Euclidean settings. One demonstration is, for example, accelerated optimization on Lie groups (Tao and Ohsawa, 2020). Following the same line of thoughts, we consider Stiefel optimization by first defining a time-dependent Lagrangian $L : T\mathsf{St} \times \mathbb{R}_+ \to \mathbb{R}$ as

$$L(X, \dot{X}, t) := r(t)\left[\frac{1}{2}g_X(\dot{X}, \dot{X}) - f(X)\right] = r(t)\left[\frac{1}{2}\mathrm{Tr}\left(\dot{X}^\top (I - aXX^\top)\dot{X}\right) - f(X)\right]. \qquad (3)$$

Without $r(t)$, this would be a standard Lagrangian for mechanics on manifold, and it would generate dynamics in which the kinetic energy $\frac{1}{2}g_X(\dot{X}, \dot{X})$ and the potential energy $f(X)$ continuously exchange with each other. Choosing $r(t)$ to be an *increasing* function, however, will break time translational symmetry and introduce dissipation in the system; consequently, the total energy $\frac{1}{2}g_X(\dot{X}, \dot{X}) + f(X)$ will monotonically decrease in time, leading to the minimization of $f$. Given this Lagrangian, we can define a variational problem (VP) on manifold whose corresponding stationarity condition (known as Euler-Lagrange equation) yields an ODE that optimizes $f$ in continuous time

$$\text{Constrained VP: } \delta \int_0^T L(X(t), \dot{X}(t), t)dt = 0, \ s.t. \ \big(X(t), \dot{X}(t)\big) \in T\mathsf{St}, \ \forall 0 \leq t \leq T. \qquad (4)$$

However, to obtain concrete algorithms, this constrained VP is very difficult to handle because the variation of $X(\cdot)$ has to keep it in a nonlinearly constrained function space. Tao and Ohsawa (2020) applied a tool of reduction via Lie algebra to solve the case of Lie group manifold, but for Stiefel manifold $\mathsf{St}(n, m)$, unless $n = m$, there is no group structure and this technique no longer applies. Therefore, we transfer (4) into an unconstrained variational problem, using tools described in, e.g., Chen et al. (2021). By adding a Lagrange multiplier[4] $\Lambda(t) \in \mathbb{R}^{m \times m}$ for the constraints $X(t)^\top X(t) - I = 0$, we augment the Lagrangian to be

$$\hat{L}(X, \dot{X}, \Lambda, t) = r(t)\left[\frac{1}{2}\mathrm{Tr}\left(\dot{X}^\top (I - aXX^\top)\dot{X}\right) - f(X)\right] - \frac{1}{2}\mathrm{Tr}\left(\Lambda^\top (X^\top X - I)\right), \qquad (5)$$

---

[3]The problem is harder when $n > m$, which thus will be our focus. For the $n = m$ case, see Sec. O.

[4]It differs from the seminal work (Wen and Yin, 2013) as our setup (and the Lagrange multiplier) is dynamical.

where $\Lambda(t) \in \mathbb{R}^{m \times m}$ is a symmetric matrix. The benefit is that now $X$ can be varied in an unconstrained, flat function space, corresponding to

$$\text{Unconstrained VP: } \delta \int_0^T \hat{L}(X(t), \dot{X}(t), \Lambda(t), t) dt = 0, \ \forall\, 0 \le t \le T. \tag{6}$$

To sum up the above discussion, problem (1) can be resolved through (un)constrained variational problem (4),(6), i.e., (1) $\impliedby$ (4) $\iff$ (6), in the sense that the solutions of (4),(6) solve problem (1); meanwhile, (6) can be explicitly solved, as detailed in the following (proof is in Apdx. G):

**Theorem 1** (Optimization dynamics on $T$St). To use problem (6) for solving problem (1), we have:

1. The solution of the unconstrained variational problem (6) is the following ODE

$$\begin{cases} \dot{X} = & Q \\ \dot{Q} = & -\gamma Q - XQ^\top Q - \frac{3a}{2}(I - XX^\top)QQ^\top X - \frac{\partial f}{\partial X} + \frac{1+b}{2}XX^\top \frac{\partial f}{\partial X} + \frac{1-b}{2}X \frac{\partial f}{\partial X}^\top X \end{cases} \tag{7}$$

   where $(X(t), Q(t)) = (X(t), \dot{X}(t)) \in \mathbb{R}^{n \times m} \times \mathbb{R}^{n \times m}$ is the tuple of state variable and its momentum, $\gamma(t) := \dot{r}(t)/r(t)$ is the scalar friction coefficient, and $b := \frac{a}{a-1}$ is a constant depending on the canonical-type metric (2).

2. For any isolated local minimum $X_* \in$ St of $f$, there exists a neighbourhood $U \subset T$St of $(X_*, \mathbf{0})$ s.t., for any initial condition $(X_0, Q_0) \in U$, the solution $(X(t), Q(t))$ of the system (7) converges to $(X_*, \mathbf{0})$ as $t \to \infty$.

One feature of ODE (7) is the preservation of constraints: even though the ODE is defined in the Euclidean space, as long as it starts on the manifold, it will never leave the manifold:

**Theorem 2** (Constrained optimization with unconstrained dynamics). If the initial condition of (7) is on $T$St, the cotangent bundle of Stiefel manifold, then dynamics (7) automatically stays on $T$St, i.e., if $X(0)^\top X(0) = I_{m \times m}$, $X(0)^\top Q(0) + Q(0)^\top X(0) = 0_{m \times m}$, then for all $t \ge 0$ $X(t)^\top X(t) = I_{m \times m}$, $X(t)^\top Q(t) + Q(t)^\top X(t) = 0_{m \times m}$. *Proof is in Apdx. H.*

## 2.2 STRUCTURE-PRESERVING DISCRETIZATION VIA VARIABLE DECOMPOSITION AND OPERATOR SPLITTING

Although the continuous dynamics (7) preserves the constraints, such preservation is in general not guaranteed when time is discretized. The construction of our discretization briefly follows four steps: a geometric decomposition of momentum, a carefully designed operator splitting scheme to approximate the ODEs, structure-preserving approximations of the split system, and a structure-preserving relaxation that further reduces the computational cost. Details now follow.

**Preparation: from a static decomposition of the tangent space to a decomposition of $Q$ dynamics.** To retain the preservation of geometric structures through a time discretization, we first decompose the tangent space $T_X$St into $X$ and $X^\perp$ components (see more details in Tagare (2011)); more precisely, given a tangent vector $Q$ represented by an $n \times m$ matrix, we rewrite it as $Q = XY + V$ for $Y \in \mathbb{R}^{m \times m}, V \in \mathbb{R}^{n \times m}$, and use $Y, V$ to replace $Q$. This transformation changes the constraint $X^\top Q + Q^\top X = 0$ to $\{Y^\top + Y = 0, \ X^\top V = 0\}$ instead (see Apdx E.1). Another advantage of this decomposition is that $Y, V$ naturally split the canonical-type metric (2) (see Apdx E.2).

Although a fixed tangent $Q$ can be uniquely decomposed into $Y$ and $V$, when we start to consider a dynamical $Q(t)$ and make the decomposition at each $t$, we need to understand how the corresponding $Y(t)$ and $V(t)$ evolve. This is not a trivial question, but it can be proved that the new dynamics is

$$\dot{X} = XY + V, \qquad \dot{Y} = -\gamma Y - \frac{1-b}{2}\left(X^\top \frac{\partial f}{\partial X} - \frac{\partial f}{\partial X}^\top X\right),$$

$$\dot{V} = -\gamma V + \frac{3a-2}{2}VY - XV^\top V - \left(I - XX^\top\right)\frac{\partial f}{\partial X}, \tag{8}$$

with initial condition satisfying $X(0)^\top X(0) = I, Y(0)^\top + Y(0) = 0, X(0)^\top V(0) = 0$. This system is equivalent to (7) via $Q(t) = X(t)Y(t) + V(t)$ and preserving $X(t)^\top X(t) = I, Y(t)^\top + Y(t) = 0, X(t)^\top V(t) = 0$ for all $t$ (Thm.8). Moreover, it will be amenable to a good discretization and is thus the base for constructing our optimizer.

With this decomposition defined, we can finally make the phrase 'structure preservation' precise:

**Definition 2.** *Structure preservation* means variables exactly satisfying all constraints for all time. That is, for '$XQ$'-system, $X(t)^\top X(t) = I, X(t)^\top Q(t) = 0, \forall t$; for '$XYV$'-system, $X(t)^\top X(t) = I, Y(t)^\top + Y(t) = 0, X(t)^\top V(t) = 0, \forall t$. In comparison, staying exactly on Stiefel, i.e. $X(t)^\top X(t) = I$ is termed 'feasible' (Wen and Yin, 2013; Ablin and Peyré, 2022), but we have additional constraints due to momentum.

**Step I: operator splitting of the ODEs.** To handle the high nonlinearity of ODE (8) and maintain the preservation of constraints after discretization, we adopt an operator splitting method, based on a general fact that a numerical discretization of an ODE can be obtained by composing the (approximate) flow maps of split ODEs (McLachlan and Quispel, 2002). More precisely, we split the vector field of (8) as a sum of three vector fields, each associated with one of the following ODEs:

$$\begin{cases} \dot{X} = XY \\ \dot{Y} = -\gamma Y \\ \quad - \frac{1-b}{2}\left(X^\top \frac{\partial f}{\partial X} - \frac{\partial f}{\partial X}^\top X\right) \\ \dot{V} = 0 \end{cases} \quad (9) \qquad \begin{cases} \dot{X} = 0 \\ \dot{Y} = 0 \\ \dot{V} = -\gamma V + \frac{3a-2}{2} VY \\ \quad -(I - XX^\top)\frac{\partial f}{\partial X} \end{cases} \quad (10) \qquad \begin{cases} \dot{X} = V \\ \dot{Y} = 0 \\ \dot{V} = -XV^\top V \end{cases} \quad (11).$$

Define the corresponding time-$h$ evolution maps $\phi_1, \phi_2, \phi_3$ of Eq.(9)(10)(11) to be $\phi_j$ : $[X(t), Y(t), V(t)] \mapsto [X(t+h), Y(t+h), V(t+h)]$ for system $j = 1, 2, 3$. Note $\phi_1, \phi_2, \phi_3$ give the exact solutions of these split ODEs. Then we see our specific split honors all constraints:

**Theorem 3.** $\phi_1, \phi_2, \phi_3$ are all structure preserving. *Proof is in Apdx. J.*

**Step II: structure-preserving approximation of exact flow maps.** Due to the nonlinearity, $\phi_1$ and $\phi_3$ do not admit analytical expressions; $\phi_2$ does have an explicit expression (Eq. (30)), but an approximation will still reduce computational costs while maintaining certain accuracy (see Fig. 4). Therefore we first denote the 1st-order approximation of the exact flow maps $\phi_1, \phi_2, \phi_3$ to be $\bar{\phi}_1, \bar{\phi}_2, \bar{\phi}_3$, where $\bar{\phi}_j : [X_0, Y_0, V_0] \mapsto [X_h, Y_h, V_h] = \phi_j([X_0, Y_0, V_0]) + \mathcal{O}(h^2), j = 1, 2, 3$. Then

$$\bar{\phi}_1 : \begin{cases} X_h = X_0 \operatorname{expm}(hY_h) \\ Y_h = \exp(-\gamma h)Y_0 \\ \quad - \frac{(1-b)(1-\exp(-\gamma h))}{2\gamma}\left(X_0^\top \frac{\partial f}{\partial X_0} - \frac{\partial f}{\partial X_0}^\top X_0\right) \\ V_h = V_0 \end{cases} \quad (12) \qquad \bar{\phi}_3 : \begin{cases} X_\dagger = X_0 + hV_0 X_0^\top X_0 \\ X_h = X_\dagger (X_\dagger^\top X_\dagger)^{-1/2} \\ Y_h = Y_0 \\ V_h = V_0 - hX_0 V_0^\top V_0. \end{cases} \quad (13)$$

$$\bar{\phi}_2 : \; X_h = X_0, \; V_h = (1-\gamma h)V_0 + \frac{3a-2}{2}hV_0 Y_0 - h\left(I - X_0 X_0^\top\right)\frac{\partial f}{\partial X}(X_0), \; Y_h = Y_0. \quad (14)$$

**Theorem 4.** $\bar{\phi}_1, \bar{\phi}_2, \bar{\phi}_3$ are all structure preserving. *Proof is in Apdx. K.*

This shows not only the split flow maps but their decently designed discretizations maintain the constraints of $X, Y, V$. There are several specially designed features to enable the numerical stability of the integrators: 1) '$X_\dagger(X_\dagger^\top X_\dagger)^{-1/2}$' in $\bar{\phi}_3$ is part of a nontrivial discretization scheme and the same as polar retraction (Absil et al., 2009). It directly leads to the preservation of the geometry of the position variable by $\bar{\phi}_3$, i.e., even the input of $\bar{\phi}_3$ has error that $X_0^\top X_0 \neq I$, the output always satisfies $X_h^\top X_h = I$, but it will not impair the order of the $\mathcal{O}(h^2)$ local discretization error. See more about its connection to numerically stability to arithmetic errors in Apdx. M.1. 2) In the first equation of $\bar{\phi}_3$, $X_0 + hV_0 X_0^\top X_0$ is used instead of $X_0 + hV_0$, even though most of the time $X_0^\top X_0 = I$. This guarantees that even if $X_0^\top X_0 \neq I$, the constraint $X_0^\top V_0 = 0$ itself still leads to $X_h^\top V_h = 0$, which improves numerical stability. What's more, when combined with 1), this property will also enable us to use a cheaper $\tilde{\phi}_1$ in the following Step III. 3) The 'forward Euler'-like discretization for $\phi_3$ is carefully chosen, and updating $V_h$ using $X_h$ instead of $X_0$, for example, will destroy its structure preservation. 4) No extra handling except forward Euler is applied to the momentum variables $Y, V$; i.e. this discretization itself is beneficial enough to guarantee momentum structure preservation.

**Step III: relaxation and composition of the operators.** Our goal is to obtain a structure preserving scheme for the original ODE (8) instead of requiring structure preservation of each operator for the split ODEs (9)(10)(11). The latter will ensure the former as we have:

**Theorem 5.** The composition of any ordering of $\bar{\phi}_1, \bar{\phi}_2, \bar{\phi}_3$, e.g., $\bar{\phi}_1 \circ \bar{\phi}_2 \circ \bar{\phi}_3$, is a structure-preserving, 1st-order (in $h$) numerical integrator of (8). *Proof is in Apdx. L.*

However, the latter (all operators preserving structure) is not needed for the former (their composition preserves structure). We can still eventually obtain a structure-preserving integrator, however without the costly 'expm', by relaxing some of the structure preservation requirements for $\bar{\phi}_1$.

**Theorem 6.** Consider a consistent approximation of $\bar{\phi}_1$ by $\tilde{\phi}_1$ (i.e. $\tilde{\phi}_1 = \bar{\phi}_1 + \mathcal{O}(h^2)$). Assume $\tilde{\phi}_1$ satisfies that if initially $X_0^\top X_0 = I, Y_0^\top + Y_0 = 0, X_0^\top V_0 = 0$, then $Y_h^\top + Y_h = 0, X_h^\top V_h = 0$. Then the specific composition $\bar{\phi}_3 \circ \tilde{\phi}_1 \circ \bar{\phi}_2$ is structure preserving. Moreover, the composition of any ordering of $\tilde{\phi}_1, \bar{\phi}_2, \bar{\phi}_3$ is a 1st-order (in $h$) integrator of (8). *Proof is in Apdx. M.3*

Therefore, our **default recommendation** is to use $\bar{\phi}_3 \circ \tilde{\phi}_1 \circ \bar{\phi}_2$, where $\tilde{\phi}_1$ is $X_h = X_0 + hX_0 Y_h$, $Y_h = (1-\gamma h)Y_0 - \frac{1-b}{2}h\left(X_0^\top \frac{\partial f}{\partial X_0} - \frac{\partial f}{\partial X_0}^\top X_0\right)$, $V_h = V_0$, mainly approximating the expm in $\bar{\phi}_1$

by forward Euler. The result is concretely summarized in Algo. 1. In order to match commonly used notations in machine learning, we rescale the parameters: learning rate $\eta := \frac{1-\exp(-\gamma h)}{\gamma} h$, momentum parameter $\mu := \exp(-\gamma h)$, $Z := Y / \frac{1-\exp(-\gamma h)}{\gamma}$, and $U := V / \frac{1-\exp(-\gamma h)}{\gamma}$. See Apdx M.4 for more information.

A few side remarks are listed in the following. If $m$ is small, then $\bar{\phi}_1 \circ \phi_2 \circ \bar{\phi}_3$ (or any permutation) can also be used, although experimentally no clear advantage was observed (Fig. 4); otherwise, the computational cost of expm can become prohibitive.

Moreover, the computational cost of matrix inverse square root in the polar retraction $(X_\dagger^\top X_\dagger)^{-1/2}$ should not be a concern since it is only for $m \times m$ matrices instead of $n \times n$ ($n \geq m$). Meanwhile, it can be computed to machine precision rapidly using a quadratically convergent iterative method (Higham, 1997); see Algo. 3 in Apdx. M.1 for details, and Apdx.C for computational complexity.

---

**Algorithm 1:** Momentum (S)GD on $\mathsf{St}(n,m)$ (SGD: $\partial f / \partial X$ replaced by a stochastic estimator)

**Hyperparameter :** $\eta \in (0, +\infty)$, $\mu \in [0,1)$, maximum number of iterations $N$
**Initialization** : $X_0, U_0, Z_0$ s.t. $X_0^\top X_0 = I$, $X_0^\top U_0 = 0$, $Z_0 + Z_0^\top = 0$

1 **for** $i = 0, \cdots, N-1$ **do**

2     Compute 'gradients': $f_i = \frac{1-b}{2}\left(X_i^\top \frac{\partial f}{\partial X}(X_i) - \frac{\partial f}{\partial X}^\top(X_i)X_i\right)$; $g_i = (I - X_i X_i^\top)\frac{\partial f}{\partial X}(X_i)$

3     Update $\bar{\phi}_2$: $U_{i+\frac{1}{2}} = \mu U_i - \frac{3a-2}{2}\eta U_i Z_i - g_i$

4     Update $\tilde{\phi}_1$: $Z_{i+1} = \mu Z_i - f_i$; $X_{i+\frac{1}{2}} = X_i + \eta X_i Z_i$

5     Update $\bar{\phi}_3$: $X_\dagger = X_{i+\frac{1}{2}} + \eta U_{i+\frac{1}{2}} X_{i+\frac{1}{2}}^\top X_{i+\frac{1}{2}}$; Compute $(X_\dagger^\top X_\dagger)^{-\frac{1}{2}}$ using Algo. 3;
      $X_{i+1} = X_\dagger (X_\dagger^\top X_\dagger)^{-\frac{1}{2}}$; $U_{i+1} = U_{i+\frac{1}{2}} - \eta X_{i+\frac{1}{2}} U_{i+\frac{1}{2}}^\top U_{i+\frac{1}{2}}$

6 **end**

7 **return** $X_N$

---

### 2.3 AN ADAPTIVE LEARNING RATE VERSION

Tuning for the best SGD learning rate can be labor intensive and computationally unaffordable, and sometimes SGD even performs worse than adaptive methods (Zhang et al., 2020). Thus in this section, we propose an adaptive version of our Stiefel optimizer. More precisely, we will establish, as an example, a Stiefel version of Adam (Kingma and Ba, 2015), which estimates the 1st and 2nd moments of gradients to obtain element-wise adaptive step sizes.

The algorithm is established via the following ideas. The 'gradient' in this Adam-version method is constructed from our Stiefel SGD with momentum (Alg.1) where the 'gradients' in $Y/Z$ and $V/U$ direction can be interpreted as $\frac{1-b}{2}\left(X^\top \frac{\partial f}{\partial X} - \frac{\partial f}{\partial X}^\top X\right)$ and $(I - X(X^\top X)^{-1}X^\top)\frac{\partial f}{\partial X}$ respectively. The main difficulty of extending Stiefel SGD to Stiefel Adam that does not appear in Euclidean case is that element-wise operation on momentum loses tangent vector structure. We solve this respectively: (1) For $Y/Z$-direction, the skew-symmetry is preserved after a symmetric element-wise operation $Z \oslash (p^{\circ\frac{1}{2}} + \epsilon)$. (2) For $V/U$-direction, we apply a projection $I - X(X^\top X)^{-1}X^\top$ to the element-wisely rescaled momentum $U \oslash (q^{\circ\frac{1}{2}} + \epsilon)$, making sure '$X^\top V = 0$'. Combining all the above, we obtain the Adam-Stiefel optimizer. Denote $\hat{\phi}_1, \hat{\phi}_2, \hat{\phi}_3$ to be the modification of $\phi_1, \phi_2, \phi_3$ in the Adam version (see detailed expressions in Apdx. M.5). Then the integrator is defined as $\hat{\phi}_3 \circ \hat{\phi}_1 \circ \hat{\phi}_2$. The overall method is shown in Algo. 2.

**Theorem 7.** The Adam-Stiefel $\hat{\phi}_3 \circ \hat{\phi}_1 \circ \hat{\phi}_2$ is structure-preserving.      *Proof is in Apdx. N.*

## 3 EXPERIMENTS

This section demonstrates our Stiefel optimizers on Projection Robust Wasserstein Distance (Paty and Cuturi, 2019; Lin et al., 2020) and trained-from-scratch Vision Transformer (Dosovitskiy et al., 2020). They will also be compared with other popular Stiefel optimizers summarized in Tab. 2. Canonical metric (i.e. $a = 1/2$ in Eq.2) is used in both examples to show that the gained performance is due to algorithmic innovation but not an extra tuned knob. An additional experiment on leading eigenvalue problem is deferred to Apdx. Q, where we also compare the convergence rates and time consumptions of different algorithms, test various choices of metrics, and study why our retraction is tailored to our algorithm. Good performance is observed in all three examples.

---

**Algorithm 2:** Adam on Stiefel manifold $\mathsf{St}(n, m)$

---

**Hyperparameter :** $\eta \in (0, +\infty)$, $\beta_1 \in [0, 1)$, $\beta_2 \in [0, 1)$, $0 < \epsilon \ll 1$, number of iterations $N$
**Initialization** $\quad$ : $X_0, V_0, Y_0, p_0, q_0$ s.t. $X_0^\top X_0 = I$, $X_0^\top U_0 = 0$, $Z_0 + Z_0^\top = 0$, $p_0 = p_0^\top$

1 **for** $i = 0, \cdots, N - 1$ **do**

2 $\quad$ Compute 'gradients': $f_i = \frac{1-b}{2}\left(X_i^\top \frac{\partial f}{\partial X}(X_i) - \frac{\partial f}{\partial X}(X_i)^\top X_i\right)$; $g_i = (I - X_i X_i^\top)\frac{\partial f}{\partial X}(X_i)$

3 $\quad$ 2nd-moment estimation: $p_{i+1} = \beta_2 p_i + (1 - \beta_2)f_i^{\circ 2}$; $q_{i+1} = \beta_2 q_i + (1 - \beta_2)g_i^{\circ 2}$

4 $\quad$ Update $\hat{\phi}_2$: $U_{i+\frac{1}{2}} = \beta_1 U_i - \frac{3a-2}{2}\eta U_i Z_i - (1 - \beta_1)g_i$

5 $\quad$ Update $\hat{\phi}_1$: $Z_{i+1} = \beta_1 Z_i - (1 - \beta_1)f_i$;
$\quad\quad X_{i+\frac{1}{2}} = X_i + \eta\sqrt{1 - \beta_2^{i+1}}X_i\left(Z_{i+1} \oslash (p_{i+1}^{\circ\frac{1}{2}} + \epsilon)\right)$

6 $\quad$ Update $\hat{\phi}_3$: $\tilde{U} = \sqrt{1 - \beta_2^{i+1}}(I - X_{i+\frac{1}{2}}(X_{i+\frac{1}{2}}^\top X_{i+\frac{1}{2}})^{-1}X_{i+\frac{1}{2}}^\top)(U_{i+\frac{1}{2}} \oslash (q_{i+1}^{\circ\frac{1}{2}} + \epsilon))$;
$\quad\quad X_\dagger = X_{i+\frac{1}{2}} + \eta\tilde{U}X_{i+\frac{1}{2}}^\top X_{i+\frac{1}{2}}$; $X_{i+1} = X_\dagger(X_\dagger^\top X_\dagger)^{-\frac{1}{2}}$; $U_{i+1} = U_{i+\frac{1}{2}} - \eta X_{i+\frac{1}{2}}\tilde{U}^\top U_{i+\frac{1}{2}}$

7 **end**

8 **return** $X_N$

---

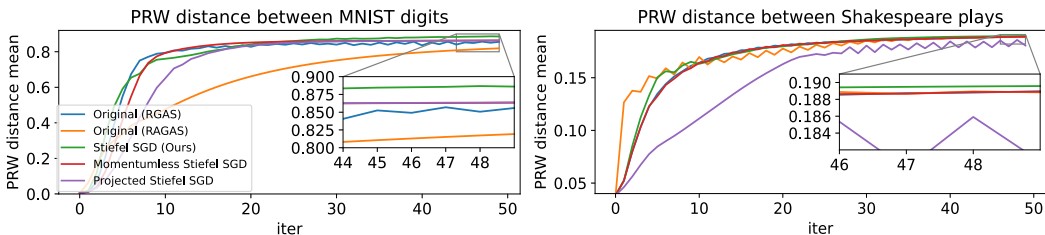

Figure 1: Projection Robust Wasserstein Distance (PRW) tested on MNIST and Shakespeare plays. Data points are features extracted by a pre-trained model. The mean optimal transport value is taken among all digits or movie pairs; larger mean optimal transport value means more effective orthogonal projection. Our method makes PRW more effective by getting the best local minimum (largest optimal transport value) and fast convergence.

More experimental details are in Apdx. P.

### 3.1 PROJECTION ROBUST WASSERSTEIN DISTANCE

Projection Robust Wasserstein Distance (PRW) (Paty and Cuturi, 2019; Lin et al., 2020) is a notion recently proposed to improve the robustness of standard Wasserstein metric, especially in high-dim settings. The idea is to simultaneously look for a best projection from high-dim data $(x_i, y_j,$ respectively with weights $r_i, c_j)$ to low-dim ones, and an entropic regularized optimal transport plan between projected data, i.e.

$$\max_{U \in \mathsf{St}(d,k)} \quad \min_{\pi \in \mathbb{R}_+^{n \times n}, \sum_j \pi_{ij} = r_i, \sum_i \pi_{ij} = c_j} \sum_{i=1}^{n}\sum_{j=1}^{n} \pi_{i,j}\|U^\top x_i - U^\top y_j\|^2 + \eta\langle \pi, \log(\pi) - 1_n 1_n^\top\rangle$$

This problem is geodesically-nonconvex w.r.t. the Stiefel variable $U$(Jordan et al., 2022), thus computationally extra-challenging to Stiefel optimizers. Lin et al. (2020) proposed an effective method based on alternations between a full Sinkhorn step (Cuturi, 2013), given the current projection, and an optimization step that improves the projection. In particular, they developed Riemmanian optimizer with projection and retraction (RGAS) and its adaptive learning rate version (RAGAS). We replace RGAS and RAGAS by our optimizer and others, and test on the hardest experiments in Lin et al. (2020), namely MNIST and Shakespeare.

Results are in Fig.1. Our method is observed to find the largest value of PRW and thus the best projection among the tested, which implies the best performance. Details of setup are in Apdx. P.2.

### 3.2 HOW COULD ORTHOGONALITY IMPROVE VANILLA VISION TRANSFORMER (VIT)?

This section explores the possibility of making self-attention in Transformer models (Vaswani et al., 2017) orthogonal. The intuition is, if we interpret each attention head as a characterization of interactions between tokens, maybe linearly dependent interactions shouldn't be double-counted so that the characterization can be more efficient and robust. In addition, there is unlikely just one way in which tokens relate to each other, and multiple heads handle different characterizations in parallel. So our second question is, should different heads also be orthogonal to each other? To formulate precisely, we follow the notation of Vaswani et al. (2017) and only review relevant parts: a Scaled Dot-Product Multihead Attention is given by $\text{MultiHead}(Q, K, V) = \text{Concat}(\text{head}_1, ..., \text{head}_{n_{\text{head}}})W^O$, where

| ViT trained from scratch (6.3M parameters) test error (mean $\pm$ std of 5 tests, in percentage) | | | |
|---|---|---|---|
| | Method | CIFAR 10 | CIFAR 100 |
| Orthogonality across heads | **Stiefel SGD** (ours, same as Tao and Ohsawa (2020) in this case) | $8.85 \pm 0.35$ | $31.51 \pm 0.39$ |
| | Projected Stiefel SGD(Li et al., 2020) | $9.65 \pm 0.23$ | $32.14 \pm 0.48$ |
| | Regularizer SGD | $9.58 \pm 0.21$ | $32.77 \pm 0.23$ |
| | **Stiefel Adam** (ours) | $8.81 \pm 0.22$ | $32.48 \pm 0.24$ |
| | Projected Stiefel Adam(Li et al., 2020) | $11.58 \pm 0.24$ | $33.98 \pm 0.12$ |
| | Regularizer Adam | $11.09 \pm 0.26$ | $33.90 \pm 0.24$ |
| Orthogonality only within each head | **Stiefel SGD** (ours) | $\underline{8.32} \pm 0.39$ | $\underline{30.20} \pm 0.55$ |
| | Projected Stiefel SGD (Li et al., 2020) | $8.90 \pm 0.18$ | $31.71 \pm 0.26$ |
| | Regularizer SGD | $9.78 \pm 0.14$ | $33.08 \pm 0.42$ |
| | Momentumless Stiefel SGD (Wen and Yin, 2013) $^{\diamond}$ | $12.90 \pm 0.22$ | $34.84 \pm 0.24$ |
| | **Stiefel Adam** (ours) | $8.46 \pm 0.20$ | $31.04 \pm 0.35$ |
| | Projected Stiefel Adam (Li et al., 2020) | $10.77 \pm 0.30$ | $33.73 \pm 0.54$ |
| | Regularizer Adam | $11.53 \pm 0.28$ | $34.72 \pm 0.67$ |
| No constraints (baseline) | SGD | $9.75 \pm 0.37$ | $32.61 \pm 0.30$ |
| | Adam | $9.52 \pm 0.32$ | $33.15 \pm 0.50$ |
| | AdamW(Loshchilov and Hutter, 2017) | $9.05 \pm 0.43$ | $33.11 \pm 0.47$ |
| Advanced methods with known error | PVT-T (12.8M parameters) Wang et al. (2021) | $9.49$ | $30.38$ |
| | DeiT-T (5.3M parameters) Touvron et al. (2021) | $11.62$ | $32.48$ |

$\diamond$: only tested for 'within each head' because it becomes ortho. SGD (Tao and Ohsawa, 2020) with 0 momentum for 'across head'.

Table 1: Orthogonal Vision Transformer (Stiefel-ViT) trained from scratch for CIFAR. For each of the 4 classes ({across, only within}$\times${SGD,Adam}), blue is best within that class. Underlined is the best over all classes, and bold faced are methods from this paper.

$\text{head}_i = \text{Attention}(QW_i^Q, KW_i^K, VW_i^V)$, $\text{Attention}(\tilde{Q}, \tilde{K}, \tilde{V}) = \text{softmax}(\frac{\tilde{Q}\tilde{K}^{\top}}{\sqrt{d_k}})\tilde{V}$, matrices $W_i^Q \in \mathbb{R}^{d_{\text{model}} \times d_k}$, $W_i^K \in \mathbb{R}^{d_{\text{model}} \times d_k}$, $W_i^V \in \mathbb{R}^{d_{\text{model}} \times d_v}$ and $W^O \in \mathbb{R}^{n_{\text{head}} d_v \times d_{\text{model}}}$ correspond to trainable parameters. The three input matrices $Q$, $K$ and $V$ all have dimension $sequence\_length \times d_{\text{model}}$. $d_k$ and $d_v$ are usually smaller than $d_{\text{model}}$.

For orthogonality only **within** head, we require that $W_i^Q$, $W_i^K$ are in $\text{St}(d_{\text{model}}, d_k)$. This needs $d_{\text{model}} \geq d_k$, which holds in most cases. For orthogonality **across** heads, we need $d_{\text{model}} \geq n_{\text{head}} d_k$, which is satisfied in many popular models, and require $\text{Concat}(W_i^Q, i = 1..., n_{\text{head}})$, $\text{Concat}(W_i^K, i = 1..., n_{\text{head}})$ to be in $\text{St}(d_{\text{model}}, n_{\text{head}} d_k)$, which means it contains not only 'orthogonality only **within** head', but also extra cross-head orthogonality.

We test these two types of constraints on vanilla ViT (Dosovitskiy et al., 2020), trained from scratch. Results summarized in Tab. 1 (and Fig. 2 in Apdx. P.3) show: (1) requiring each head to be orthogonal just within itself leads to the most significant improvement of performance; (2) additionally requiring heads to be orthogonal to each other will actually be less effective, although that is still better than requiring no orthogonality; (3) the choice of optimizer matters: our methods give the best results and out-perform non-constrained baselines in all cases, but not all existing Stiefel optimizers can beat non-constrained baselines in all cases; (4) momentum significantly helps train orthogonal ViT; (5) simply imposing orthogonal constraint (with our optimizer) makes ViT outperform some carefully designed models of the same size that are trained from scratch (Tab. A3 in Zhang et al. (2022)). Note models trained from scratch are very different from pre-trained ViT. To have some realistic expectation of the performance of trained-from-scratch ViT, recall, for example, an improved ViT, PVT-T (Wang et al., 2021), has 9.49% and 30.38% error on CIFAR 10 and CIFAR 100, using 12.8M parameters, and another improvement, DeiT-T (Touvron et al., 2021), achieves 11.62% and 32.48% respectively with 5.3M parameters. Our model in Tab.1 (and Fig.2 in Apdx. P.3) uses 6.3M parameters. More details about this experiment can be found in Apdx. P.3.

**Remark 1** (No additional hyperparameters needed for the Stiefel part). Our Stiefel-ViT model has both non-Euclidean and Euclidean parameters, the latter due to the reason that we don't impose orthogonality on feedforward layers and $W_i^V$. Our optimizer can use the same learning rate for both sets of parameters. This is because the optimizer is the time discretization of ODEs, and synchronous discretization is sufficient for convergence. This contrasts with other approaches such as projected Steifel SGD/Adam (Li et al., 2020), where learning rates for these two sets of parameters need to be adjusted separately and could differ by 40 times in some tasks. See Apdx. P.1 for more discussion.

## ACKNOWLEDGMENTS

We thank Tomoki Ohsawa for insightful discussion. We are grateful for partial support by NSF DMS-1847802 (LK, YW and MT), NSF ECCS-1936776 (MT), and Cullen-Peck Scholar Award (LK, YW and MT).

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

**Notation** For simplicity, we denote $G := \frac{\partial f}{\partial X}$ when there is no confusion.

# A  A QUICK SUMMARY OF OPTIMIZERS WE EXPERIMENTALLY COMPARED TO IN THIS PAPER

| manifold preserving | Optimizer | Pros and Cons |
|---|---|---|
| ✓ | Stiefel SGD/Adam (Ours) | ✓ allows a family of metrics
✓ fast iterative method for position retraction
✓ momentum needs no retraction
✓ learning rate can be made adaptive
✓ same learning rate for Stiefel & Euclidean parameters |
| ✓ | Projected Stiefel SGD/Adam (Li et al., 2020) | ✓ learning rate can be made adaptive
✗ slower iterative method for position retraction; fast if early stopped but then manifold preservation is lost
✗ momentum needs projection every iteration
✗ different learning rates for Stiefel & Euclidean parameters |
| ✓ | Momentumless Stiefel (S)GD (Wen and Yin, 2013) | ✗ no momentum |
| ✗ | Regularizer (Cisse et al., 2017) | ✓ fast computation
✗ sensitive to regularization strength |

Table 2: A summary of pros and cons of existing optimizers.

# B  TECHNICAL REMARKS ON GEOMETRIC MECHANICS

## B.1  INTUITION OF THE COTANGENT BUNDLE

If we require exact Stiefel manifold preservation for all time, i.e., $X(t)^\top X(t) = I$, a simple time differentiation gives a 2nd-order constraint $X(t)^\top Q(t) + Q(t)^\top X(t) = 0$ for $Q = \dot{X}$. This is why momentum $Q$ has the additional structure $Q(t) \in T^*_{X(t)}\mathsf{St}$.

## B.2  WHAT EXACTLY IS MOMENTUM?

Through this paper, we abused notation and called $Q := \dot{X}$ momentum, following the common practice of the community. It really should be called velocity instead, because although in canonical Euclidean spaces this doesn't make much a difference, for mechanical systems on manifolds they are not the same thing. For example, for our case, the geometrically intrinsic momentum variable should be given by Legendre transform

$$P := \frac{\partial L}{\partial \dot{X}} = r(I - aXX^\top)\dot{X} \tag{15}$$

instead, and velocity $Q$ is in the tangent space while momentum $P$ is in the cotangent space. In the paper we used the identity map for an isomorphism between the tangent and cotangent spaces, but if we'd like to make our variational formulation more elegant by viewing the kinetic energy as a natural pairing between the velocity and momentum variables, then the isomorphism should be given by the metric as in (15).

None of these affect the correctness or the efficacy of results in this paper. This discussion is just about terminology.

## B.3  THE SPECIAL CASE OF $\mathsf{St}(n, n)$ AND ITS RELATION WITH $\mathsf{O}(n)$ AND $\mathsf{SO}(n)$

When $n \geq m$, $\mathsf{St}(n, m) \cong \mathsf{O}(n)/\mathsf{O}(n - m)$; when $n > m$ strictly, we also have $\mathsf{St}(n, m) \cong \mathsf{SO}(n)/\mathsf{SO}(n - m)$. However, in the special case of $n = m$, $\mathsf{St}(n, n) \cong \mathsf{O}(n)$ is not connected

unlike the $n > m$ cases. Since our optimizer is based on the discretization of continuous dynamics, it cannot make jumps and thus just optimizes on a connected component of $\mathsf{St}(n, m)$, which means for the special case of $n = m$, it is, to be precise, optimizing on $\mathsf{SO}(n)$ but not $\mathsf{O}(n)$, although a similar complication is nonexistent when $n > m$.

## C   DETAILS ABOUT THE PER-ITERATION COMPLEXITY AND COMPUTATIONAL COST FOR OUR ALGORITHMS

The most costly operation in our algorithms is the $n \times m$ matrix multiplication (note $n > m$). The computation for the matrix exponential and square root of matrix inversion, is cautiously designed to only deal with matrices of dimension $m \times m$ (see Apdx. M.1M.2), and thus admits at most $\mathcal{O}(m^3)$ at each step (particularly, forward Euler only has the complexity of $\mathcal{O}(m^2)$ while Cayley map is $\mathcal{O}(m^3)$). For the inner loop of the square root of matrix inversion (Algo. 3), due to its quadratic convergence (Higham, 1997), it takes $\mathcal{O}(\log \log(1/\mathbf{u}))$ number of steps to achieve the machine precision $\mathbf{u}$. Note early stopping can be applied to this loop so that the complexity can be further reduced while the order of the overall method (1st-order) is still maintained. The combination of all the above gives complexity $\mathcal{O}(nm^2) + \mathcal{O}(m^3 \log \log(1/\mathbf{u}))$[5].

Commonly used optimizers on Stiefel manifold (e.g. Tab. 2) also have per-iteration complexity of $\mathcal{O}(nm^2)$. Although it is difficult to compare them with our method in terms of the constant prefactors in respective complexity bounds, we can do some heuristics to estimate these prefactors, which will demonstrate the low-cost advantage of our method. For our `Stiefel SGD` (Algo. 1), we count the numbers of matrix multiplications needed per iteration, each of which costs $nm^2$. Namely, step 2: 2 multiplications; step 3: 1; step 4: 1; step 5: 6. In step 6 we need to call Algo. 3 for computing matrix root inversion, and a closer look gives 3 $m^3$-cost matrix multiplications in each iteration of Algo. 3. Since Fig. 5(c) shows that 8 inner iteration is enough for this inner loop to converge under double precision, in total, 10 $nm^2$-cost matrix multiplications and 24 $m^3$-cost matrix multiplications are needed in each (outer) iteration of our optimizer (Algo. 1). We also count the number of matrix multiplications needed for other aforementioned algorithms. For `Momentumless Stiefel (S)GD` (Wen and Yin, 2013), the smartly designed Cayley map retraction takes about 10 times of $nm^2$-cost matrix multiplications and one inversion for a $(2m) \times (2m)$-sized matrix in each iteration. In comparison, `Projected Stiefel SGD` Li et al. (2020) needs 9 $nm^2$-cost matrix multiplications for projecting momentum and 6 $nm^2$-cost matrix multiplications each iteration of Cayley loop. Since 8 iterations are needed to compute the Cayley transform to single precision (see fig. 5; note it is just single precision instead of double precision by our inner loop), a total of 57 $nm^2$-cost matrix multiplications are needed per outer iteration.

The above estimation is just for the part on maintaining the structure of the position variable. Let's now also discuss the momentum variable. Our carefully designed way of introducing momentum bypasses the costly moving momentum while keeping it in tangent spaces. Note `Projected Stiefel SGD` first moves momentum in the Euclidean space and then uses a cleverly designed projection, which markedly improves computational efficiency, but it still devotes 9 matrix multiplications to just the momentum projections, let alone the fact that needing projection means less accuracy. To add to this discussion, parallel transport is another existing way of moving momentum, but it cannot even be solved with $\mathcal{O}(nm^2)$ complexity.

Altogether, the above estimations heuristically show that our prefactor is close to that in `Momentumless Stiefel (S)GD` but much smaller than those in existing methods that have momentum. This matches the experimental evidence in Fig. 3.

## D   TERMINOLOGIES OF RIEMANNIAN OPTIMIZATION

In each iteration of optimization, gradient needs to be a vector in tangent space, and the operation that maps nonintrinsic gradient to the tangent space is called projection. Then, the variable goes one step on manifold given the tangent vector, ideally via the exponential map, which is, however, expensive or even not admitting a closed form computation, and approximation known as retraction is used.

---

[5]$\mathbf{u}$ stands for machine precision, which is a very small number

When momentum, a tangent vector, is involved, at the same time we update the position, tangent space is also changed. An intrinsic way to move tangent vectors is by parallel transport, which is an isomorphism of the tangent spaces of two points connected by a geodesic.

# E  ABOUT THE $Y, V$ DECOMPOSITION

## E.1  THE GEOMETRIC VERSION AND THE DYNAMICAL VERSION OF THEIR STRUCTURAL CONSTRAINTS

A tangent vector $Q \in T_X\mathsf{St}$ can be decomposed, as studied in Edelman et al. (1998); Absil et al. (2009), into

$$Q = \begin{pmatrix} X & X^\perp \end{pmatrix} \begin{pmatrix} Y \\ Z \end{pmatrix} = XY + X^\perp Z, \tag{16}$$

where $X \in \mathsf{St}(n, m), Y \in \mathbb{R}^{m \times m}, Z \in \mathbb{R}^{(n-m) \times m}; X^\perp \in \mathbb{R}^{n \times (n-m)}$ is a matrix in the orthogonal complement of $X$, i.e., $X^\top \cdot X^\perp = \mathbf{0}_{m \times (n-m)}$. Let $V = X^\perp Z$. We have $X \perp V$ and

$$\begin{cases} Y = X^\top Q \\ V = Q - XY = (I - XX^\top)Q. \end{cases} \tag{17}$$

The above representation immediately implies $Y$ is a skew-symmetric matrix since

$$X^\top Q + Q^\top X = 0 \Rightarrow Y^\top = -Y. \tag{18}$$

In our derivation, however, $X$ and $Q$ are variables (they really should be $X(t) \in \mathsf{St}, Q(t) \in T_{X(t)}\mathsf{St}$) that change with time, and this poses a new challenge. More precisely, given the $Q$ dynamics, i.e. $\dot{Q}$, there could be infinitely many ways of assigning $\dot{Y}$ and $\dot{V}$ so that they together produce the given $\dot{Q}$; however, in general they do not maintain the tangent decomposition structures. More precisely, starting with

$$Y(0)^\top + Y(0) = 0 \text{ and } X(0)^\top V(0) = 0$$

(and hence $X(0)^\top Q(0) + Q(0)^\top X(0) = 0$), one may not have

$$Y(t)^\top + Y(t) = 0 \text{ and } X(t)^\top V(t) = 0 \text{ for } t > 0$$

despite that $X(t)^\top Q(t) + Q(t)^\top X(t) = 0$ will still be guaranteed. However, we have found a nontrivial choice of $\dot{Y}$ and $\dot{V}$ (Eq.8), so that the (static) geometric constraint becomes dynamically true, i.e. $Y(t)^\top + Y(t) = 0$ and $X(t)^\top V(t) = 0$ for all $t$. Therefore we can simply maintain the decomposition

$$\begin{cases} Y(t) = X(t)^\top Q(t) \\ V(t) = Q - XY = (I - X(t)X(t)^\top)Q(t). \end{cases}$$

## E.2  THE ADVANTAGE OF THE $Y, V$ DECOMPOSITION

The $Y, V$ decomposition is near 'orthogonal' in the sense that $XY \perp V$ and it makes the metric separable. More precisely, consider our $Y, V$ representation. $\forall \Delta_i \in T_X St, i = 1, 2$, if we denote $Y_{\Delta,i} = X^\top \Delta_i$ and $V_{\Delta,2} = (I - XX^\top)\Delta_i$, which means $\Delta_i = XY_{\Delta,i} + V_{\Delta,i}$, then the metric (2) can be rewritten as

$$\begin{aligned} g_X(\Delta_1, \Delta_2) &= \mathrm{Tr}((XY_{\Delta,1} + V_{\Delta,1})^\top (I - aXX^\top)(XY_{\Delta,2} + V_{\Delta,2})) \\ &= (1 - a)\mathrm{Tr}(Y_{\Delta,1}^\top Y_{\Delta,2}) + \mathrm{Tr}(V_{\Delta,1}^\top V_{\Delta,2}). \end{aligned}$$

This means the metric of the tangent space is a linear combination of the metrics of $Y$ and $V$-directions. Therefore, this gives an intuition why our continuous dynamics and numerical discretization in the following can handle the family of canonical-type metrics.

## F CHOICE OF $\gamma$

Although monotonicity of $r$ suffices for the convergence, choices of $\gamma = \dot{r}/r$ affect the convergence speed just like in Euclidean cases (e.g.,Wibisono et al. (2016); Wilson et al. (2021)). Popular choices include constant $\gamma$ and $\gamma(t) = 3/t$, but methods proposed here work for all $\gamma$'s (e.g., $\gamma = 3/t + ct^p$ for variance reduction Tao and Ohsawa (2020)). However, for simplicity we will use constant $\gamma$ from now on.

## G PROOF OF THEOREM 1

*Proof of Theorem 1, Part 1.* Given the Lagrangian

$$L(X, \dot{X}, \Lambda, t) = r(t) \left[ \frac{1}{2} \text{Tr} \left( \dot{X}^\top (I - aXX^\top) \dot{X} \right) - f(X) \right] - \frac{1}{2} \text{Tr} \left( \Lambda^\top (X^\top X - I) \right), \quad (19)$$

Legendre transform gives momentum $P$ as

$$P := \frac{\partial L}{\partial \dot{X}} = r(I - aXX^\top)\dot{X}.$$

One can equivalently switch the Hamiltonian picture, and the corresponding Hamiltonian is $H : T^*\mathsf{St} \to \mathbb{R}$ with $H(X, P) := Tr(P^\top \dot{X}) - L$

$$H(X, P) = \frac{1}{2r} \text{Tr} \left[ P^\top (I - bXX^\top) P \right] + rf(X) + \frac{1}{2} \text{Tr} \left[ \Lambda^\top (X^\top X - I_{m \times m}) \right] \quad (20)$$

where $b := \frac{a}{a-1}$ solves $(I - aXX^\top)(I - bXX^\top) = I$. Hence we get the following Hamilton's equations

$$\begin{cases} \dot{X} = \dfrac{\partial H}{\partial P} = \dfrac{1}{r}(I - bXX^\top)P \\ \dot{P} = -\dfrac{\partial H}{\partial X} = \dfrac{b}{2r} \left( PP^\top X + XP^\top P \right) - r\dfrac{\partial f}{\partial X} - X\Lambda \end{cases} \quad (21)$$

Since $X^\top X \equiv I$, which gives $\dot{X}^\top X + X^\top \dot{X} = 0$. So we have $X^\top P + P^\top X = 0$. Take derivative to get $\dot{X}^\top P + X^\top \dot{P} + \dot{P}^\top X + P^\top \dot{X} = 0$, and use the condition that $\Lambda$ is symmetric, we can solve $\Lambda$ is

$$\Lambda = \frac{b+2}{2r} P^\top P - \frac{b}{2r} X^\top PP^\top X - \frac{r}{2}(X^\top G + G^\top X) \quad (22)$$

And $(X, P)$ system becomes

$$\begin{cases} \dot{X} = \dfrac{\partial H}{\partial P} = \dfrac{1}{r}(I - bXX^\top)P \\ \dot{P} = -\dfrac{\partial H}{\partial X} = \dfrac{b}{2r} PP^\top X - \dfrac{1}{r} XP^\top P + \dfrac{b}{2r} XX^\top PP^\top X - rG + \dfrac{r}{2}(XX^\top G + XG^\top X) \end{cases} \quad (23)$$

By a coordinate change $Q := \frac{1}{r}(I - bXX^\top)P \in T_X\mathsf{St}$ and define friction parameter $\gamma := \dot{r}/r$, Eq. (23) becomes

$$\dot{Q} = -\frac{\dot{r}}{r^2}(I - bXX^\top)P + \frac{1}{r}(I - bXX^\top)\dot{P} - \frac{b}{r} \dot{X} X^\top P - \frac{b}{r} X \dot{X}^\top P$$

$$= -\gamma Q - XQ^\top Q - \frac{3a}{2}(I - XX^\top)QQ^\top X - G + \frac{1+b}{2} XX^\top G + \frac{1-b}{2} XG^\top X$$

$$\begin{cases} \dot{X} = Q \\ \dot{Q} = -\gamma Q - XQ^\top Q - \dfrac{3a}{2}(I - XX^\top)QQ^\top X - G + \dfrac{1+b}{2} XX^\top G + \dfrac{1-b}{2} XG^\top X \end{cases} \quad (24)$$

$\square$

In order to prove the second part of Thm. 1, we need the following lemma.

**Lemma 1** (First-order stationary point). If $X \in \mathsf{St}$ s.t. $G - \frac{1+b}{2}XX^\top G - \frac{1-b}{2}XG^\top X = 0$, then $\forall \Delta \in T_X\mathsf{St}$, we have $\mathrm{Tr}(G^\top \Delta) = 0$, which means $X$ is a first-order stationary point of $f$.

*Proof.* Left multiply both side of $G - \frac{1+b}{2}XX^\top G - \frac{1-b}{2}XG^\top X = 0$ by $XX^\top$, we have $XX^\top G = XG^\top X$, further we also have $G = XG^\top X$.

So we have

$$
\begin{aligned}
\mathrm{Tr}(G^\top \Delta) =& \mathrm{Tr}(X^\top G X^\top \Delta) \\
=& -\mathrm{Tr}(X^\top G \Delta^\top X) \\
=& -\mathrm{Tr}(\Delta^\top X X^\top G) \\
=& -\mathrm{Tr}(\Delta^\top G)
\end{aligned}
$$

Thus we have $\mathrm{Tr}(\Delta^\top G) = 0, \forall \Delta \in T_X\mathsf{St}$, which means $X$ is a 1-order stationary point of $f$.

$\square$

*Proof of Theorem 1, Part 2.* Let $t \to (X(t), Q(t))$ be a solution of Eq. (7). Define the 'energy' function $E : T\mathsf{St} \to \mathbb{R}$ as

$$
E(X, Q) := \frac{1}{2}\mathrm{Tr}(Q^\top(I - aXX^\top)Q) + f(X) \tag{25}
$$

This gives a Lyapunov function. So we have a neighbourhood $U$ of $(X_*, 0)$ such that $E(X, Q) \geq f(X) \geq f(X_*)$ for any $(X, Q) \in U$. More over, since $X^\top Q + Q^\top X \equiv 0$, we have

$$
\frac{dE}{dt}(X(t), Q(t)) = \mathrm{Tr}(Q^\top(I - aXX^\top)\dot{Q}) - a\mathrm{Tr}(Q^\top(\dot{X}X^\top)Q) + \mathrm{Tr}\left(\frac{\partial f}{\partial X}^\top \dot{X}\right) \tag{26}
$$

Using the fact that $X^\top Q + Q^\top X \equiv 0$, we have $\mathrm{Tr}(Q^\top XQ^\top Q) = 0$ and $\mathrm{Tr}(Q^\top XX^\top G) + \mathrm{Tr}(Q^\top XG^\top X) = 0$, which gives

$$
\frac{dE}{dt}(X(t), Q(t)) = -\gamma\mathrm{Tr}(Q^\top(I - aXX^\top)Q) \leq 0 \tag{27}
$$

Since we have $r$ monotonely increasing, which means $\gamma = r'/r > 0, \forall t$. Then we have the energy is decreasing monotonically, implying that $Q = 0$ when converged. By lemma 1 that the limiting point for $X(t)$ is a first order stationary point, which is $X_*$ since it is an isolated local minimum. $\square$

**Remark 2.** For the sake of length, rate of convergence in specific situations (e.g., under geodesic convexity) will not be quantified, but it should be obtainable via tools in Wilson et al. (2021); Duruisseaux and Leok (2021).

## H  PROOF OF THEOREM 2

*Proof.* We can derive a new system of ODEs as

$$
\begin{cases}
\dfrac{d}{dt}(X^\top X) = X^\top Q + Q^\top X \\[2mm]
\dfrac{d}{dt}(X^\top Q) = -\gamma X^\top Q - (I - X^\top X)Q^\top Q - \dfrac{3a}{2}X^\top(I - XX^\top)QQ^\top X \\[2mm]
\qquad\qquad\quad - X^\top G + \dfrac{1+b}{2}X^\top XX^\top G + \dfrac{1-b}{2}X^\top XG^\top X
\end{cases}
$$

with viewing $X^\top G$ as a matrix function of $t$ following Eq. (7). By the uniqueness and existence of ODE, we have that this system of ODE with variable $(X^\top X, X^\top Q)$ and initial condition $(I, 0)$ has the unique solution $X^\top X \equiv I, X^\top Q \equiv 0$. $\square$

## I  XYV SYSTEM IS STRUCTURE PRESERVING

**Theorem 8** (Constrained optimization with unconstrained dynamics). As long as the initial condition of (8) satisfies

$$X(0)^\top X(0) = I_{m \times m}, \qquad Y(0) + Y(0)^\top = 0_{m \times m}, \qquad X(0)^\top V(0) = 0_{m \times m},$$

then the dynamics automatically satisfies the same constraint, i.e.,

$$X(t)^\top X(t) = I_{m \times m}, \qquad Y(t) + Y(t)^\top = 0_{m \times m}, \qquad X(t)^\top V(t) = 0_{m \times m}, \qquad (28)$$

*Proof.* By the uniqueness of solution of ODE, we know that the following ODE for $(X^\top X, Y^\top + Y, X^\top V)$, derived from Eq. (8), has a unique solution if view $V^\top V$ as an independent variable.

$$\begin{cases} \dfrac{d}{dt}\left(X^\top X\right) = X^\top XY + Y^\top X^\top X + X^\top V + V^\top X \\[2mm] \dfrac{d}{dt}(Y^\top + Y) = -\gamma(Y^\top + Y) \\[2mm] \dfrac{d}{dt}(X^\top V) = -\gamma X^\top V + YX^\top V + \dfrac{3a-2}{2}X^\top VY + (I - X^\top X)V^\top V \end{cases} \qquad (29)$$

We can see that given initial condition $(X_0^\top X_0, Y_0^\top + Y_0, X_0^\top V_0) = (I, 0, 0)$, the unique solution is $(X^\top X, Y^\top + Y, X^\top V) \equiv (I, 0, 0)$. So we know that the constraint in Eq. (28) are preserved by continuous dynamics Eq. (8). $\qquad \square$

## J  PROOF OF THEOREM 3

*Proof.* We assume the initial condition $(X_0, Y_0, V_0)$ satisfies constraint (28).

For Eq. (9), we check $X^\top(t)X(t) \equiv 0$, $Y(t) + Y^\top(t) \equiv 0$ and $X^\top(t)V(t) \equiv 0$.

$$\begin{aligned} \frac{d}{dt}(Y^\top + Y) &= \dot{Y}^\top + \dot{Y} \\ &= -\gamma(Y + Y^\top) \end{aligned}$$

with initial condition $Y_0 + Y_0^\top = 0$, we have $Y + Y^\top \equiv 0$

$$\begin{aligned} \frac{d}{dt}X^\top V &= \dot{X}^\top V + X^\top \dot{V} \\ &= Y^\top XV \end{aligned}$$

with initial condition $X_0^\top V_0 = 0$, we have $X^\top V \equiv 0$

$$\begin{aligned} \frac{d}{dt}X^\top X &= X^\top \dot{X} + \dot{X}^\top X \\ &= X^\top XY + X^\top V + Y^\top X^\top X + V^\top X \\ &= X^\top XY + Y^\top X^\top X \end{aligned}$$

Using the conclusion $Y + Y^\top \equiv 0$ that we have just proved and initial condition $X_0^\top X_0 = I$, we have $X^\top X \equiv 0$.

For Eq. (10), we have the exact solution of $\phi_2$ is given by $X(t) = X(0), Y(t) = Y(0)$, and with $M := \gamma I_{m \times m} - \frac{3a-2}{2}Y(t)$,

$$V(t) = V(0)\exp m(-Mt) - \left(I - X(0)X(0)^\top\right)\frac{\partial f}{\partial X}(X(0))M^{-1}(I - \exp m(-Mt)), \qquad (30)$$

For Eq. (11), we need to check $\frac{d}{dt}\left(X^\top(t)X(t)\right) = 0$ and $\frac{d}{dt}(X^\top(t)V(t)) = 0$.

By the uniqueness of solution of ODE, we know that the following ODE for $(X^\top X, X^\top V, V^\top V)$, derived from Eq. (11), has a unique solution

$$
\begin{cases}
\dfrac{d}{dt}\left(X^\top X\right) = X^\top V + V^\top X \\[2mm]
\dfrac{d}{dt}(X^\top V) = (I - X^\top X)V^\top V \\[2mm]
\dfrac{d}{dt}(V^\top V) = -V^\top X V^\top V - V^\top V X^\top V
\end{cases}
\tag{31}
$$

We can see that given initial condition $(X_0^\top X_0, X_0^\top V_0, V_0^\top V_0) = (I, 0, V_0^\top V_0)$, the unique solution is $(X^\top X, X^\top V, V^\top V) \equiv (I, 0, V_0^\top V_0)$. So we know that the constraint $X^\top X = I$, $X^\top V = 0$ are preserved by continuous dynamics Eq. (11).

$\square$

## K    Proof of Theorem 4

*Proof.* **Eq. (12) is structure preserving.** We use the idea from Tao and Ohsawa (2020). Assume the initial value $X_0$, $Y_0$ and $V_0$ satisfies the constraint, i.e., $X_0^\top X_0 = I$, $Y_0 + Y_0^\top = 0$ and $X_0^\top V_0 = 0$. For the first step updating $Y$, due to the special form of the derivative of $Y$ that is always skew-symmetric, we can tell that $Y_h$ is also skew-symmetric and $Y_h + Y_h^\top = 0$. And the skew-symmetricity of $Y_h$ gives us that

$$
X_h^\top X_h = \mathrm{expm}(hY_h^\top)X_0^\top X_0 \mathrm{expm}(hY_h)^\top = 0
$$
$$
X_h^\top V_h = \mathrm{expm}(hY_h^\top)X_0^\top V_0 = 0
$$

so all 3 conditions in Eq. (28) are satisfied, meaning Eq. (12) is structure preserving.

**Eq. (14) is structure preserving.**

Given initial condition $X_0^\top X_0 = I$ and $X_0^\top V_0 = 0$, we check constraint $X_h^\top V_h = 0$.

$$
X_h^\top V_h = (1 - \gamma h)X_0^\top V_0 + \frac{3a - 2}{2}hX_0^\top V_0 Y_0 - hX_0^\top\left(I - X_0 X_0^\top\right)\frac{\partial f}{\partial X}(X_0) = 0
$$

**Eq. (13) is structure preserving.** First, we show that the discretization is a one order approximation of the exact solution. We get $V_h$ and $X_\dagger$ by forward Euler. And since $X_0$ and $V_0$ satisfies the constraint 28, we have

$$
\begin{aligned}
X_\dagger^\top X_\dagger &= (X_0 + hV_0)^\top(X_0 + hV_0) \\
&= (X_0^\top X_0 + hX_0^\top V_0 + hV_0^\top X_0 + h^2 V_0^\top V_0) \\
&= I + h^2 V_0^\top V_0
\end{aligned}
$$

which means $X_h = X_\dagger + \mathcal{O}(h^2) = X_0 + hV_0 + \mathcal{O}(h^2)$, indeed a one order approximation of exact solution.

Next we show the numerical discretization is structure preserving. For constraint $X^\top V = 0$, we can check that

$$
X_\dagger^\top V_h = X_0^\top V_0 + hX_0^\top X_0 V_0^\top V_0 - hX_0^\top X_0 V_0^\top V_0 - h^2 V_0^\top X_0 V_0^\top V_0 = 0
$$

which gives us $X_h^\top V_h = 0$. For restriction $X^\top X = I$, we have

$$
X_h^\top X_h = (X_\dagger^\top X_\dagger)^{-\frac{1}{2}} X_\dagger^\top X_\dagger (X_\dagger^\top X_\dagger)^{-\frac{1}{2}} = I
$$

$\square$

## L    Proof of Theorem 5

*Proof.* The composition of structure-preserving maps is structure preserving. $\phi_1 \circ \phi_2 \circ \phi_3$ is a 1st-order integrator due to operator splitting theory McLachlan and Quispel (2002), and its convergence order is kept after any $\phi_j$ gets replaced by $\bar{\phi}_j$ as long as the difference between $\bar{\phi}_j$ and $\phi_j$ is higher-order Tao (2016). □

## M    Discussions on our numerical integrator

### M.1    Benefits of the step $X_\dagger(X_\dagger^\top X_\dagger)^{-\frac{1}{2}}$

To remain the position $X$ on the manifold, certain techniques that pull point back to manifold are used. For example, in Li et al. (2020); Lin et al. (2020) for an $n$-by-$m$ matrix $X$ that is not on the manifold, they perform $QR$ decomposition $X = QR$ and update the new $X$ to be the first $m$ columns of $Q$. However, the $QR$ decomposition is not unique and such step does not have a closed form expression. More importantly, it cannot help design a structure-preserving scheme.

Instead, our algorithm has a similarly functioned step via the square root of the inverse matrix $X_h = X_\dagger(X_\dagger^\top X_\dagger)^{-1/2}$. In this case, it follows that $X_h^\top X_h = I$. This step is carefully designed from discretization of ODE ensuring the structure of $X, Y, V$ is preserved at the same time. Through this update, we are able to obtain a structure-preserving method with a closed form expression (see the proof of Theorem 4 in Apdx. K). Note that the square root of the inverse matrix can be iteratively solved with low computational cost (see Algo. 3; Higham (1997)). It was proved to be quadratically convergent, which means only a couple of iterations are needed to reach machine precision, and only matrix multiplications are needed. This lead to our inner loop is more efficient in both cost per iteration and convergence speed comparing to Li et al. (2020). See Sec. Q for more details.

---

**Algorithm 3:** Algorithm for matrix root and matrix root inversion (Eq. (2.6) in Higham (1997))

**Input:** Symmetric $m$-by-$m$ matrix $A$, *tol*
**Initialization**        : $Y_0 = A$, $Z_0 = I_{m \times m}$, $k = 0$
1  **while** $\|Y_k^2 - A\| \geq$ *tol* **do**
2  $\quad$ $Y_{k+1} = \frac{1}{2}Y_k(3I - Z_kY_k)$
3  $\quad$ $Z_{k+1} = \frac{1}{2}(3I - Z_kY_k)Z_k$
4  $\quad$ $k \leftarrow k + 1$
5  **end**
6  **return** $Y_k \approx A^{\frac{1}{2}}$ *and* $Z_k \approx A^{-\frac{1}{2}}$

---

Another benefit of this step is that it is more stable to the truncation error produced by finite machine precision (which can become a significant issue if the model is trained in single accuracy on consumer graphic cards or even quantized). Although $X_\dagger^\top X_\dagger$ follows the expression

$$\begin{aligned} X_\dagger^\top X_\dagger &= (X_0 + hV_0)^\top(X_0 + hV_0) \\ &= (X_0^\top X_0 + hX_0^\top V_0 + hV_0^\top X_0 + h^2 V_0^\top V_0) \\ &= I + h^2 V_0^\top V_0, \end{aligned}$$

the stability of $X_\dagger(X_\dagger^\top X_\dagger)^{-1/2}$ is much better than $X_\dagger(I + h^2 V_0^\top V_0)^{-1/2}$. Particularly, when $n = m$, it is an identical map if there is no machine error.

### M.2    Simplification of matrix exponential

There are two steps in the discretizations (30) and (12) that use the matrix exponential of $m \times m$ matrices. Similar matrix exponential is also shown in Li et al. (2020); Wen and Yin (2013) but with matrices of much larger size $n \times n$ (note $n > m$). We introduce two ways of simplifying the computation of the matrix exponential. First, note that Cayley transform is a 2nd-order structure-preserving approximation of the matrix exponential, defined as follows

$$\text{Cayley}(hY) = (I - hY/2)^{-1}(I + hY/2) = \exp(hY) + \mathcal{O}(h^3). \tag{32}$$

By applying the Cayley transform, the computation is reduced to an inversion of matrix and matrix multiplication while it still keeps the variable on the manifold.

Additionally, we can use the first-order forward Euler to discretize the ODEs for the two steps, which is also the first-order truncation of the matrix exponential. In this case, the $X$ update in scheme (12) is just $X_h = X_0 + hY_h$ and is no longer structure-preserving but structure-preserving property of the overall algorithm is not affected (see Thm. 6).

For scheme (30), its Cayley map approximation is structure preserving and defined as follows

$$V_h = V_0 \, \text{Cayley}(-\gamma hI + \frac{3a-2}{2}hV_0Y_0) - \frac{1-\exp(-\gamma h)}{\gamma}(I - X_0X_0^\top)\frac{\partial f}{\partial X_0}. \qquad (33)$$

Note the variable $V$ will still satisfy the constraint (28) even if we change the update of $V$ to forward Euler

$$V_h = V_0 - \gamma hV_0 + \frac{3a-2}{2}hV_0Y_0 - \frac{1-\exp(-\gamma h)}{\gamma}(I - X_0X_0^\top)\frac{\partial f}{\partial X_0}. \qquad (34)$$

In fact, the above two ways and the original matrix exponential share the same complexity per iteration. Also, we do not find any significant difference in the convergence in numerical experiments (Apdx. Q). Hence we will always use forward Euler as default, which has the lowest computational cost.

### M.3    Proof of Theorem 6

*Proof.* The composition of $\bar{\phi}_1$, $\bar{\phi}_2$, $\bar{\phi}_3$ in any order will give a structure preserving scheme, because in Sec. K we proved that each of them is structure preserving.

When we apply it in specific order $\bar{\phi}_3 \circ \bar{\phi}_1 \circ \bar{\phi}_2$, we prove when $X^\top X = I$ is no longer preserved by $\bar{\phi}_2$, the assembled scheme is still structure preserving. We only need to prove that $[X_h, Y_h, V_h] = \bar{\phi}_3(x_0, Y_0, V_0)$ satisfies the following: when initial condition satisfies $Y_0^\top + Y_0 = 0$ and $X_0^\top V_0 = 0$, we have $X_h^\top X_h = I$, $Y_h^\top + Y_h = 0$ and $X_h^\top V_h = 0$.

Since we have the step $X_h = X_\dagger(X_\dagger^\top X_\dagger)^{-1/2}$, we have $X_h^\top X_h = I$. $Y$ is not updating so $Y_h + Y_h^\top = 0$.

$$X_\dagger^\top V_h = X_0^\top V_0 - hX_0^\top X_0V_0^\top V_0 + hX_0^\top X_0V_0^\top V_0 - h^2X_0^\top X_0V_0^\top X_0V_0^\top V_0 = 0$$

As a result, $X_h^\top V_h = (X_\dagger^\top X_\dagger)^{-1/2}X_\dagger^\top V_h = 0$

$\square$

### M.4    The integrator in rescaled coordinates used in Algorithm 1

$$\tilde{\phi}_1 : \begin{cases} X_\eta = X_0 + \eta Z_h \\ Z_\eta = \mu Z_0 - \frac{1-b}{2}\left(X_0^\top\frac{\partial f}{\partial X_0} - \frac{\partial f}{\partial X_0}^\top X_0\right) \\ U_\eta = U_0 \end{cases} \qquad \bar{\phi}_2 : \begin{cases} X_\eta = X_0 \\ Z_\eta = Z_0 \\ U_\eta = \mu U_0 + \frac{3a-2}{2}\eta U_0Z_0 \\ \qquad - (I - X_0X_0^\top)\frac{\partial f}{\partial X_0}, \end{cases}$$

$$\bar{\phi}_3 : \begin{cases} X_\dagger = X_0 + \eta U_0X_0^\top X_0 \\ X_\eta = X_\dagger(X_\dagger^\top X_\dagger)^{-1/2} \\ Z_\eta = Z_0 \\ U_\eta = U_0 - \eta X_0U_0^\top U_0 \end{cases}$$

## M.5 THE INTEGRATOR IN RESCALED COORDINATES USED IN ALGORITHM 2

'gradient' and 2-order momentum :
$$
\begin{cases}
f_i = \dfrac{1-b}{2}\left(X_i^\top \dfrac{\partial f}{\partial X}(X_i) - \dfrac{\partial f}{\partial X}(X_i)^\top X_i\right) \\[2mm]
g_i = (I - X_i X_i^\top)\dfrac{\partial f}{\partial X}(X_i) \\[2mm]
p_{i+1} = \beta_2 p_i + (1-\beta_2)f_i^{\circ 2} \\[2mm]
q_{i+1} = \beta_2 q_i + (1-\beta_2)g_i^{\circ 2}
\end{cases}
$$

$$
\hat{\phi}_1 : \begin{cases}
X_{i+1} = & X_i + \eta\sqrt{1-\beta_2^{i+1}}\,Z_{i+1}\oslash(p_{i+1}^{\circ 1/2}+\epsilon) \\
Z_{i+1} = & \beta_1 Z_i - (1-\beta_1)f_i \\
U_{i+1} = & U_i
\end{cases}
\qquad
\hat{\phi}_2 : \begin{cases}
X_{i+1} = & X_i \\
Z_{i+1} = & Z_i \\
U_{i+1} = & \beta_1 U_i - \frac{3a-2}{2}\eta U_i Z_i - (1-\beta_1)g_i
\end{cases}
$$

$$
\hat{\phi}_3 : \begin{cases}
\tilde{U} = \sqrt{1-\beta_2^{i+1}}(I - X_i(X_i^\top X_i)^{-1}X_i^\top)(U_i\oslash(q_{i+1}^{\circ 1/2}+\epsilon)) \\
X_\dagger = X_i + \eta\tilde{U}X_i^\top X_i \\
X_{i+1} = X_\dagger(X_\dagger^\top X_\dagger)^{-1/2} \\
Z_{i+1} = Z_i \\
U_{i+1} = U_i - \eta X_i\tilde{U}^\top U_i
\end{cases}
$$

## N PROOF OF THEOREM 7

*Proof.* We will use the same notation as in algorithm 2. Assume $X_i^\top X_i = I$, $Y_i^\top + Y_i = 0$, $X_i^\top V_i = 0$, $p_i = p_i^\top$. Our initialization satisfies these conditions. We prove that these conditions are satisfied after mapped by $\hat{\phi}_3 \circ \hat{\phi}_1 \circ \hat{\phi}_2$ in the following. **Step 1**: Due to the skew-symmetricity of $f_i$, we have $p_{i+1}$ is symmetric.

**Step 2**: $X_i, Z_i, U_{i+\frac{1}{2}} := \hat{\phi}_2(X_i, Z_i, U_i)$, where $U_{i+\frac{1}{2}} = \beta_1 U_i - \frac{3a-2}{2}\eta U_i Z_i - (1-\beta_1)g_i$. Since $X_i^\top U_i = 0$ and $X_i^\top g_i = 0$, we can check

$$
X_i^\top U_{i+\frac{1}{2}} = \beta_1 X_i^\top U_i - \frac{3a-2}{2}\eta X_i^\top U_i Z_i - (1-\beta_1)X_i^\top g_i = 0
$$

**Step 3**: $X_{i+\frac{1}{2}}, Z_{i+1}, U_{i+\frac{1}{2}} := \hat{\phi}_1(X_i, Z_i, U_{i+\frac{1}{2}})$, where $Z_{i+1} = \beta_1 Z_i - (1-\beta_1)f_i$ and $X_{i+\frac{1}{2}} = X_i + \eta\sqrt{1-\beta_2^{i+1}}X_i\left(Z_{i+1}\oslash(p_{i+1}^{\circ\frac{1}{2}}+\epsilon)\right)$.

Since we have $X_i^\top U_{i+\frac{1}{2}} = 0$ We can have directly $X_{i+\frac{1}{2}}^\top U_{i+\frac{1}{2}} = 0$. And due to $f_i$ is symmetric, $Z_{i+1}$ is still skew-symmetric. Note that $X_{i+\frac{1}{2}}^\top X_{i+\frac{1}{2}} = I$ no longer stands now but it satisfied until the algorithm finishes.

**Step 4** $X_{i+1}, Z_{i+1}, U_{i+1} := \hat{\phi}_3(X_{i+\frac{1}{2}}, Z_{i+1}, U_{i+\frac{1}{2}})$. First we can see from $X_{i+1} = X_\dagger(X_\dagger^\top X_\dagger)^{-\frac{1}{2}}$ that $X_{i+1}^\top X_{i+1} = I$ as long as $X_\dagger = X_{i+\frac{1}{2}} + \eta\tilde{U}X_{i+\frac{1}{2}}^\top X_{i+\frac{1}{2}}$ is full rank, which is always true since $X_i$ is full rank and $\eta$ is small.

Since we have $X_{i+\frac{1}{2}}^\top \tilde{U} = 0$ by the special form of matrix $I - X_{i+\frac{1}{2}}(X_{i+\frac{1}{2}}^\top X_{i+\frac{1}{2}})^{-1}X_{i+\frac{1}{2}}^\top$

$$
X_\dagger^\top U_{i+\frac{1}{2}} = \eta X_{i+\frac{1}{2}}^\top X_{i+\frac{1}{2}}\tilde{U}^\top U_{i+\frac{1}{2}}
$$

Since $X_\dagger^\top X_\dagger = I$,

$$
X_{i+1}^\top U_{i+1} = (X_\dagger^\top X_\dagger)^{-\frac{1}{2}}X_{i+\frac{1}{2}}^\top\left(U_{i+\frac{1}{2}} - \eta X_{i+\frac{1}{2}}\tilde{U}^\top U_{i+\frac{1}{2}}\right) = 0
$$

So we proved all the constrain are satisfied again, which means $X_{i+1}^\top X_{i+1} = I$, $Y_{i+1}^\top + Y_{i+1} = 0$, $X_{i+1}^\top V_{i+1} = 0$, $p_{i+1} = p_{i+1}^\top$ □

## O   OPTIMIZATION ON $\mathsf{SO}(n)$ AS A SPECIAL CASE FOR $n = m$ ON $\mathsf{St}(n, m)$

Our method for Stiefel optimization can naturally be applied on the special orthogonal group $\mathsf{SO}(n)$ (Apdx. B.3) and is defined as follows

$$\mathsf{SO}(n) := \{X \in \mathbb{R}^{n \times n} : X^\top X = I_n, \det X = 1\}. \tag{35}$$

Tao and Ohsawa (2020) proposed an efficient algorithm based on Lie group structure for the optimization on $\mathsf{SO}(n)$ although it cannot be generalized to Stiefel case. Our method restores the same integrator in Tao and Ohsawa (2020) while using different approach applied to a family of metrics.

In greater detail, for $\mathsf{SO}(n)$, the canonical-type metric (2) degenerates to Euclidean metric up to constant scaling, i.e., $g(\Delta_1, \Delta_2) = (1 - a)Tr(\Delta_1^\top \Delta_2)$. Then the corresponding Lagrangian is

$$L(X, \dot{X}, t) = r(t)\left[\frac{1-a}{2}\text{Tr}\left(\dot{X}^\top \dot{X}\right) - f(X)\right] - \frac{1}{2}\text{Tr}\left(\Lambda^\top (X^\top X - I)\right), \tag{36}$$

which results in the following position-momentum $(X, Q)$ dynamics

$$\begin{cases} \dot{X}(t) = Q(t) \\ \dot{Q}(t) = -\gamma(t)Q(t) - X(t)Q(t)^\top Q(t) \\ \qquad - \dfrac{\partial f}{\partial X}(t) + \dfrac{1+b}{2}X(t)X^\top(t)\dfrac{\partial f}{\partial X}(t) + \dfrac{1-b}{2}X(t)\dfrac{\partial f}{\partial X}(t)^\top X(t) \end{cases} \tag{37}$$

with the same initialization as (7). Next, apply the same $Y, V$ decomposition of $Q$. Since $I - XX^\top = 0$, we have $V = 0$, i.e., the momentum $Q$ is purely in $Y$-direction. Hence the equivalent dynamics to (37) is the following

$$\begin{cases} \dot{X}(t) = X(t)Y(t) \\ \dot{Y}(t) = -\gamma Y(t) - \dfrac{1-b}{2}\left(X^\top(t)\dfrac{\partial f}{\partial X}(t) - \dfrac{\partial f}{\partial X}^\top(t)X(t)\right). \end{cases} \tag{38}$$

Note when we take $b = -1$ (namely, $a = 1/2$), (38) is identical to the continuous dynamics in Tao and Ohsawa (2020), although Tao and Ohsawa (2020) uses an intrinsic form which is coordinate-free but not explicit enough.

The numerical integrator corresponding to (S)GD is defined the same as Section M.4 and is summarized in Alg.4. Note Step 5 is optional in the $n = m$ case. It gives the identity map if no arithmetic error due to machine precision is incurred. However, this step is beneficial under low precision (see Apdx. M.1).

---

**Algorithm 4:** Momentum SGD on $\mathsf{SO}(n)$ (generalization to Tao and Ohsawa (2020))

---

**Hyperparameter :** $\eta \in (0, +\infty)$, $\gamma \in [0, +\infty)$, $a < 1$, maximum number of iteration $N$
**Initialization**     : $X_0, V_0, Y_0$ s.t. $X_0^\top X_0 = I$, $X_0^\top V_0 = 0$, $Y_0 + Y_0^\top = 0$, $b = \frac{a}{a-1}$
**Output:** Local minimum $X$ of $f$

1 **for** $i = 0, ..., N - 1$ **do**
2 $\quad$ $f_i = \frac{1-b}{2}\left(X_i^\top \frac{\partial f}{\partial X}(X_i) - \frac{\partial f}{\partial X}(X_i)^\top X_i\right)$
3 $\quad$ $Y_{i+1} = \mu Y_i - f_i$
4 $\quad$ $X_\dagger = X_i \text{expm}(\eta Y_{i+1})$
5 $\quad$ $X_{i+1} = X_\dagger (X_\dagger^\top X_\dagger)^{-\frac{1}{2}}$
6 **end**
7 **return** $X_N$

---

In addition, we also have an adaptive version which was absent in Tao and Ohsawa (2020):

### O.1   ADAM ON $\mathsf{SO}(n)$

In this section, we extend the above optimizer to an Adam version. Note it can be derived from either the special case of the Stiefel optimizer, or the structure of Lie algebra $\mathfrak{so}(n)$, i.e., left-trivialization

(see Tao and Ohsawa (2020)) such that the momentum $Y$ can be recognized as an element in $T_I\mathsf{SO}(n)$, the tangent space at the identity. In the latter interpretation, the momentum will always stay in the same tangent space $T_I\mathsf{SO}(n)$ which passes on a message that trivialization almost reproduce the convenience in Euclidean space for $\mathsf{SO}(n)$. Hence unlike the general Adam-Stiefel optimizer (Algo. 2), no extra technique, for example projection, is needed for Adam-$\mathsf{SO}(n)$. Detailed iterations are shown in Algo. 5. Note this is also a structure preserving scheme (see Thm. 7).

---

**Algorithm 5:** Adam on $\mathsf{SO}(n)$

---

**Hyperparameter :** $\eta \in (0, +\infty)$, $\beta_1 \in [0,1)$, $\beta_2 \in [0,1)$, $0 < \epsilon \ll 1$, maximum number of iteration $N$
**Initialization** : $X_0, V_0, Y_0$ s.t. $X_0^\top X_0 = I$, $X_0^\top V_0 = 0$, $Y_0 + Y_0^\top = 0$, $p_0 = 0$, $q_0 = 0$

1 **for** $i = 0, ..., N-1$ **do**

2 $\quad f_i = \frac{1-b}{2}\left(X_i^\top \frac{\partial f}{\partial X}(X_i) - \frac{\partial f}{\partial X}(X_i)^\top X_i\right)$

3 $\quad p_{i+1} = \beta_2 p_i + (1-\beta_2)f_i^{\circ 2}$

4 $\quad Y_{i+1} = \beta_1 Y_i - (1-\beta_1)f_i$

5 $\quad X_\dagger = X_i \text{expm}\left(\eta\sqrt{1-\beta_2^{i+1}}Y_{i+1} \oslash (p_{i+1}^{\circ-\frac{1}{2}} + \epsilon)\right)$

6 $\quad X_{i+1} = X_\dagger(X_\dagger^\top X_\dagger)^{-\frac{1}{2}}$.

7 **end**

8 **return** $X_N$

---

## P  EXPERIMENTAL DETAILS

Note: codes are provided in supplementary materials.

Experiments are conducted on a high-performance computing cluster whose name shall be revealed post anonymous period. Single v100 GPU was used.

### P.1  GENERAL DISCUSSION

**Initialization**   For variable $X$ with the initial $X_0$, in fact, any initialization with full rank is valid, including the non-orthogonal $X_0$ such that $X_0$ does not start on the manifold. This is due to the structure-preserving property of our algorithm (both SGD and Adam versions, see Apdx. M). After just one iteration, $X$ will stay on the manifold. Here we suggest one way of obtaining orthogonal $X_0$ which is performed in Saxe et al. (2013). After randomly generating an entry-wise i.i.d. normal distributed $n$-by-$m$ matrix, we can perform the QR decomposition. Denote the orthogonal matrix from QR decomposition as $\tilde{Q}$ and denote the diagonal matrix which is the sign of the diagonal elements of $R$ as $\tilde{R}$. Then we can take $X_0 = \tilde{Q}\tilde{R}$ and thus have $X_0^\top X_0 = I$.

For the variables $U, Z$ in Stiefel SGD and $U, Z, p, q$ in Stiefel Adam, we would use zero matrices as their initialization since zero matrices always satisfy these constraints (28).

**No need to tune learning rate separately for constrained and unconstrained parameters**   For problems with both constrained and unconstrained parameters (Stiefel and Euclidean spaces), for example, the ViT test in Sec. 3, existing literatures using projection, retraction, and intermediate matrices (see Sec. 1.1) requires separate adjustments of learning rates for the two types of parameters. As an illustration, in projected SGD and Adam on Stiefel manifold Li et al. (2020), learning rate may vary 10s of times for constrained and unconstrained parameters. This can be intuitively understood as the result of applying different optimization methods for different parameters. However, using various learning rates may lead to the divergence of the algorithms.

In contrast, our Stiefel method can be established for both the constrained and unconstrained parameters with the same learning rate. Due to the same derivation of numerical methods for these two types, the effort in tuning hyperparameters is reduced while there is still good performance. In practice, the learning rate need to be adjusted in Stiefel SGD and momentum can be chosen as 0.9 for most machine learning tasks. For Stiefel Adam, we recommend to use $10^{-3}$ to be the learning rate and $(\beta_1, \beta_2) = (0.9, 0.999)$.

| Method | MNIST | Shakespear |
|---|---|---|
| Original (RGAS) | $\eta = 8e - 4$ | $\eta = 2$ |
| Original (RAGAS) | $\eta = 0.01, \beta = 0.8$ | $\eta = 0.08, \beta = 0.9$ |
| Stiefel SGD (ours)) | $\eta = 1e - 3, \mu = 0.5$ | $\eta = 2, \mu = 0.5$ |
| Projected Stiefel SGDLi et al. (2020) | $\eta = 7e - 4, \mu = 0.5$ | $\eta = 15, \mu = 0.5$ |
| Momentumless Stiefel SGD Wen and Yin (2013) | $\eta = 6e - 4$ | $\eta = 2$ |

Table 3: Details for hyperparameters of optimizers in PRW test. $\eta$ stands for learning rate, $\mu$ stands for momentum parameter and $\beta$ stands for the parameter that adjusts stepsize automatically.

**How to choose between Adam-Stiefel and momentum SGD-Stiefel**    Generally, we recommend choosing momentum SGD-Stiefel for traditional optimization and small scale neural networks that hyperparameters can be carefully adjusted while for large scale neural networks, use Adam-Stiefel as default.

### P.2   DETAILS OF PRW

For MNIST experiment, we use a pretrained CNN to extract 128-dim. features of figures in MNIST. For Shakespeare's plays experiment, we compute the PRW distances between all pairs of items in a corpus of eight Shakespeare's operas embedded into 200-dim. space using word2vec. For each digit or play pair, we use the extracted features as point sets $\{x_i\}$ and $\{y_i\}$. We choose the dimension of the target space of the projection to be $k = 2$ in both experiments. The mean optimal transport values are taken among all digits or movie pairs at the specific iteration. All the setting are same as the original paper Lin et al. (2020) except that early termination is removed.[6] The hyperparameters for each optimizer are listed in Tab. 3.

### P.3   DETAILS OF ViT EXPERIMENTS

The structure of ViT and training process is exactly same as `https://github.com/omihub777/ViT-CIFAR` but we use a different ViT implementation from `https://github.com/lucidrains/vit-pytorch`. The hyperparameters are carefully adjusted for each optimizer to obtain the best performance. For projected Stiefel SGD/Adam Li et al. (2020), we tune the learning rates for constrained and unconstrained parameters separately. For regularizer SGD/Adam, the regularizer scaling parameter is also carefully chosen. The largest learning rate for momentumless Stiefel SGD Wen and Yin (2013) is applied but momentum methods still admit faster convergence. Additional experimental results can be seen in Fig. 2.

**Model structure**    The images are cut into $4 \times 4$ patches. $d_{model} = d_{feedforward} = 384, n_{head} = 12$. $d_q = d_k = d_v = 32$ (which also equals $d_{model}/n_{head}$, so 'orthogonality across heads' constraint can be applied; see Section 3.2). The 'classification token' in Dosovitskiy et al. (2020) is used. Total number of layer of our ViT model is 7.

**Data augmentation**    Following `https://github.com/omihub777/ViT-CIFAR`, we use the implementation of (Cubuk et al., 2018) from `https://github.com/DeepVoltaire/AutoAugment` .

**Training**    All the training uses the same scheduler, which is to let the learning rate linearly increase from 0 to the target learning rate in 5 epochs, followed by cosine annealing learning rate Loshchilov and Hutter (2016) with no restart, and minimum learning rate as 0.01 times max learning rate. Label smoothing with parameter 0.1 is used. weight decay $= 5e - 5$ unless specified.

All the hyperparameters used for optimizers are listed in Tab. 4.

---

[6]Adam-type methods are not included since they are not suitable for this problem.

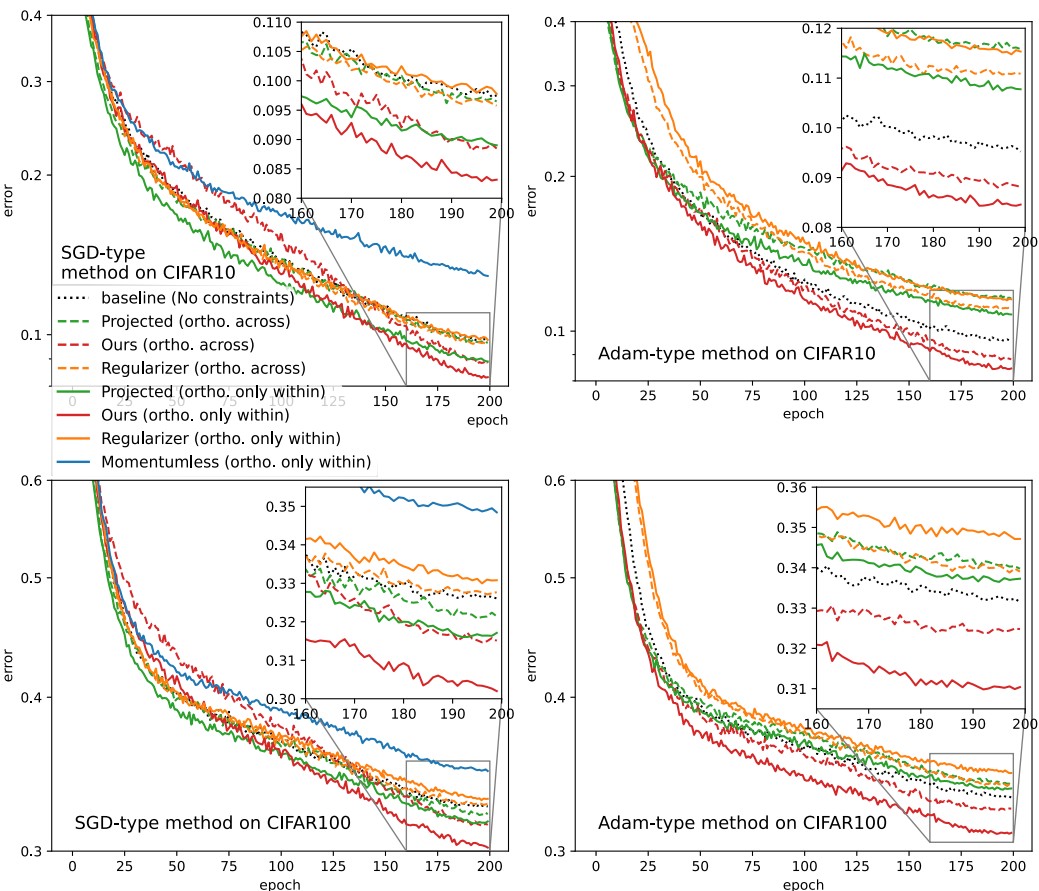

Figure 2: Test errors of Stiefel-ViT on CIFAR. We can see that our optimizer has the best test accuracy in both cases. Meanwhile, when our optimizer is used, orthogonality only within each head performs better than orthogonality across heads.

| Method | Hyperparameters |
|---|---|
| Stiefel SGD (Algo. 1)) | $\eta_{orth} = \eta_{non-orth} = 0.15$, $\mu = 0.9$ |
| Projected Stiefel SGD(Li et al., 2020) | $\eta_{orth} = 1$, $\eta_{non-orth} = 0.1$, $\mu = 0.9$ |
| Regularizer SGD | $\eta = 0.1$, $\mu = 0.9$, penalty weight = 1e-6 |
| Momentumless Stiefel SGD (Wen and Yin, 2013) | $\eta_{orth} = 0.1$. For non-orth parameters, $\eta_{non-orth} = 0.1$, $\mu = 0.9$ |
| Stiefel Adam (Algo. 2) | $\eta_{orth} = \eta_{non-orth} = 0.001$, $\beta_1 = 0.9$, $\beta_2 = 0.999$ |
| Projected Stiefel Adam(Li et al., 2020) | $\eta_{orth} = 0.001$, $\eta_{non-orth} = 0.01$, $\beta_1 = 0.9$, $\beta_2 = 0.999$ |
| Regularizer Adam | $\eta = 0.001$, $\beta_1 = 0.9$, $\beta_2 = 0.999$, penalty weight = 1e-6 |
| SGD | $\eta_{orth} = 0.1$, $\mu = 0.9$ |
| Adam (Kingma and Ba, 2015) | $\eta = 0.001$, $\beta_1 = 0.9$, $\beta_2 = 0.999$ |
| AdamW (Loshchilov and Hutter, 2017) | $\eta = 0.001$, $\beta_1 = 0.9$, $\beta_2 = 0.999$, weight decay $= 0.2$ |

Table 4: Details for hyperparameters of optimizers in ViT training. $\eta$ stands for learning rate, $\mu$ stands for momentum parameter and $\beta_1$, $\beta_2$ are parameters in Adam.

| Method | Hyperparameters |
|---|---|
| Stiefel SGD (ours) | $\eta = 0.1$, $\mu = 0.9$ |
| Stiefel Adam (ours) | $\eta = 0.001$, $\beta_1 = 0.9$, $\beta_2 = 0.999$ |
| Projected Stiefel SGDLi et al. (2020) | $\eta = 0.2$, $\mu = 0.9$ |
| Momentumless Stiefel SGD Wen and Yin (2013) | $\eta = 0.1$ |

Table 5: Details for hyperparameters of optimizers in leading eigenvalue problem. $\eta$ stands for learning rate, $\mu$ stands for momentum parameter and $\beta_1$ and $\beta_2$ stands for the hyperparameters in Adam.

## Q    ADDITIONAL EXPERIMENT: SYSTEMATIC TESTS ON THE LEADING EIGENVALUE PROBLEM

In this section, we consider the leading eigenvalue problem: given an $n$-by-$n$ matrix $A$, find the $m$ largest eigenvalues of $A$. This problem is at the core of many data sciences tasks, where $n$ can be very large but $m$ remains small. Due to its relative simplicity and the existence of exact solution, it's possible to implement a large amount of experiments in order to systematically investigate the convergence, manifold perseverance, and time consumption of various approaches. Particularly, this problem can be formulated as an optimization problem on $\mathsf{St}(n, m)$ as follows

$$\arg \max_{X \in \mathsf{St}(n,m)} \mathrm{Tr}(X^\top A X). \tag{39}$$

The idea is one seeks an optimal $m$-dimensional subspace, represented by $X$ via $m$ orthonormal bases in $\mathbb{R}^n$ corresponding to its $m$ columns, so that eigenvalues projected onto this subspace sums up to a maximum value.

Such formulation is not unique. For example, Tao and Ohsawa (2020) consider applying their momentum-accelerated $\mathsf{SO}(n)$ optimizer to $\arg \min_{R \in \mathsf{SO}(n)} \mathrm{Tr}(E^\top R^\top A R E)$, where the constant padding matrix $E := [I_{m \times m}; 0_{(n-m) \times m}]$. However, their setting is computationally less efficient than our Stiefel simplification. In fact, our formulation will be particularly suitable to cases when $m$ is small but $n$ is very large.

Our test uses a deterministic matrix $A$ generated by $A = (\Xi + \Xi^\top)/2/\sqrt{n}$, where $\Xi$ is an instance of $n$-by-$n$ matrix with i.i.d. random normal elements. The exact solution is obtainable from matrix decomposition so that the error can be quantified. As is mentioned above, several aspects of our algorithm are examined, including convergence speed, manifold perseverance, and computational complexity in Fig. 3, as well as other exact manifold-preserving methods. We can conclude that our (S)GD-Stiefel method has the fastest convergence rate and exact manifold perseverance with the lowest computational complexity.

Additionally, in Fig. 4, we test the influence of different canonical-type metric, i.e., different $a$ in Def. 1, and the three different ways to 'compute' matrix exponential in Apdx. M.2, i.e., the matrix exponential itself, Cayley map, and forward Euler. No significant difference is found in leading eigenvalue test and as a result, the most commonly used canonical metric and the cheapest forward Euler are chosen as default in all the other experiments. The hyperparameter used for each algorithm are listed in Tab. 5

We also experimentally prove that better manifold preservation leads to better convergence, for both our algorithm 1 and Projected Stiefel SGD (Li et al., 2020) in Fig 5. What's more, this experiment shows our (inner) iterative solver is quadratically convergent (as opposed to the linearly convergent Cayley approximation in Li et al. (2020)), and therefore a smaller number of iteration is needed to reach machine precision.

---

[7]The time recorded excludes the corresponding part of computing gradients.

Both the algorithm in paper Li et al. (2020) and their code has $\mathcal{O}(n^2 m)$ computational complexity per iteration. However, this can be improved by changing the order of computing matrix production by associative law. Also, this $\mathcal{O}(n^2 m)$ is the complexity under a fixed number of iterations for approximating Cayley map according to their setup. It will be changed to $\mathcal{O}(nm^2(1 + \log(1/\mathbf{u})))$ if Cayley map is computed to machine precision.

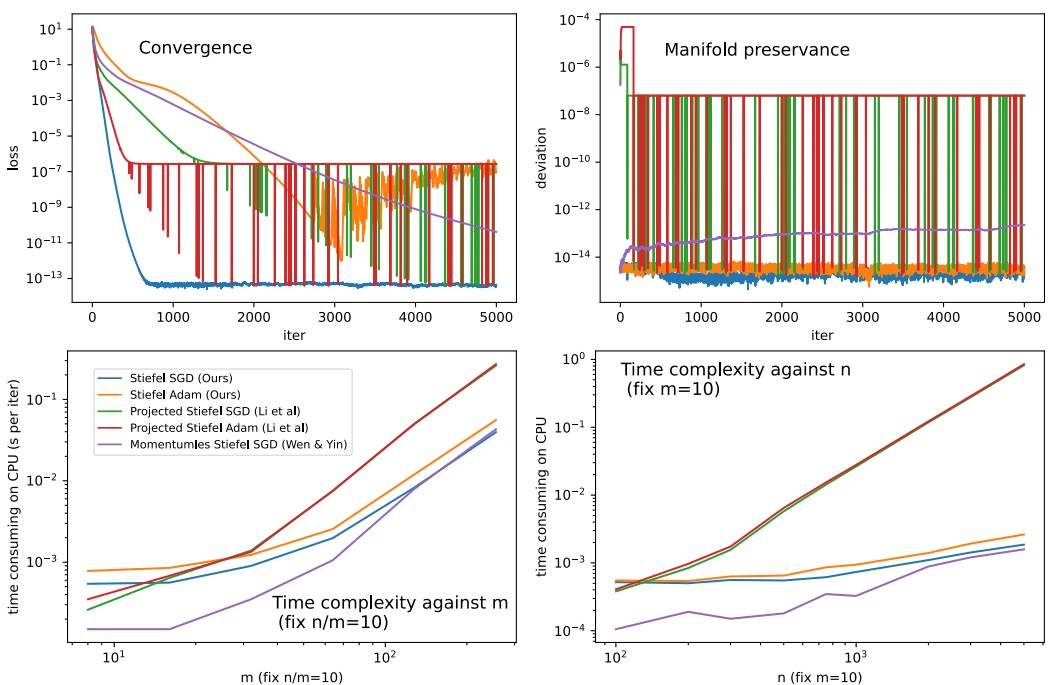

Figure 3: Comparison of exact manifold preserving methods on leading eigenvalue problem. The algorithms are performed on CPU with learning rate of each one adjusted to its best performance. The upper two figures show that our Stiefel SGD optimizer has the fastest convergence and best manifold perseverance. The lower two figures experimentally prove that we have the lowest $\mathcal{O}(nm^2)$ complexity per iteration, which matches our complexity analysis in Apdx. C and table 2. Notice that our algorithm and `Momentumless Stiefel SGD` have almost coinciding curves when $m \to \infty$ in (c) and when $n \to \infty$ in (d), while both algorithms have much lower curves than `Projected Steifel SGD` when $m \to \infty$. This means that our algorithm has a similar constant factor as the complexity of `Momentumless Stiefel (S)GD` (Wen and Yin, 2013), indicating that the introduction of momentum in our algorithm is almost 'free', and meanwhile, `Projected Steifel SGD` (Li et al., 2020) has a much larger prefactor. Please see Apdx. C for more discussions. [7]

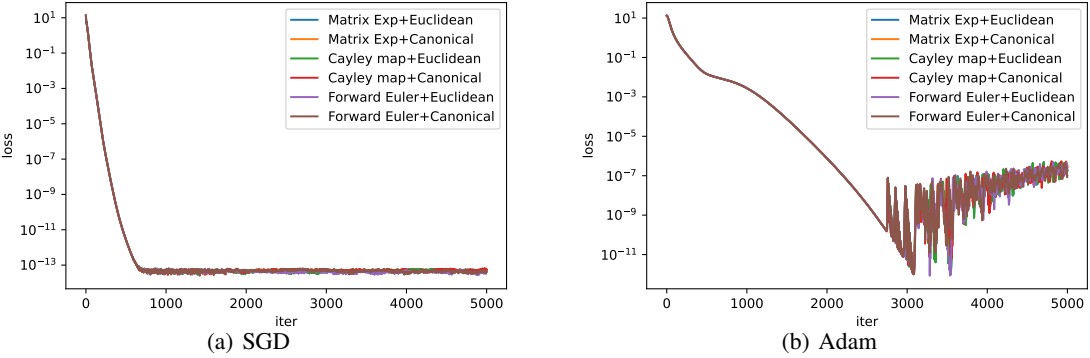

Figure 4: Comparison of different canonical-type metrics (Euclidean and canonical metrics are tested) and different ways to approximate matrix exponential in Apdx. M.2.

Though our special retraction on tangent bundle (polar retraction for position and no extra handling for momentum) is cheap, it must be used with our specially designed algorithm and cannot be applied to other algorithms even if the position will still stay on Stiefel manifold. The reason is that the loss of structure of momentum will lead to slow convergence. Fig 6 shows that projected Stiefel SGD with our retraction for tangent bundle convergences slower than not only our algorithm but also the original projected Stiefel SGD.

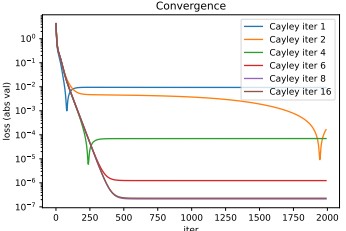

(a) Projected Stiefel SGD Li et al. (2020) using different numbers of Cayley loops

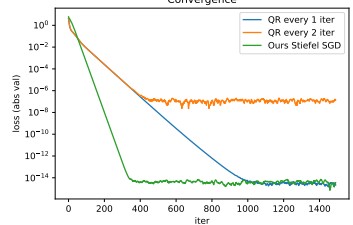

(b) Projected Stiefel SGD Li et al. (2020) using QR retractions in different frequencies

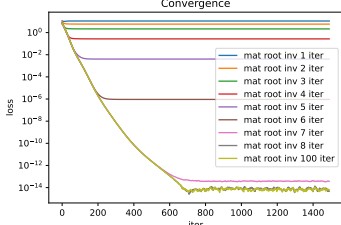

(c) Algo. 4 using different number of inner loops for matrix root inversion

Figure 5: (a) shows that on Projected Stiefel SGD, performing more inner iterations that leads to a more accurate Cayley transform gives a better optimized function value; (b) shows that more frequently applying QR retractions, i.e., projecting the variable, provides better accuracy; (c) shows that more accurate matrix root inversion for polar retraction in our algorithm 4 leads to more accurate solution. We can see performing 8 iterations (note this curve coincides with 100 iterations) of matrix root inversion gives perfect accuracy for our method, compared to (a) where 16 iterations of Cayley transform only give an error $\sim 10^6$ times bigger.

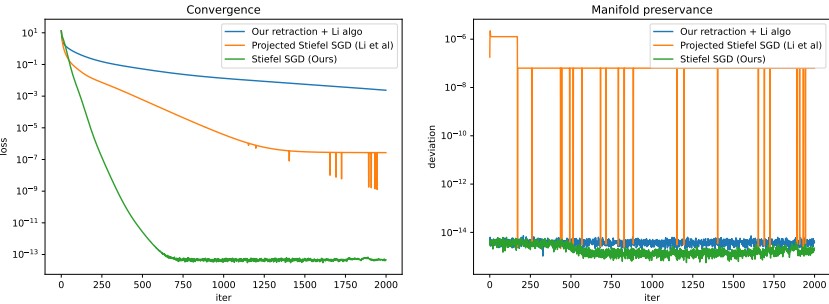

Figure 6: Convergence and manifold preservation when our low-cost retraction (polar retraction for position and no extra handling for momentum) is applied to projected Stiefel SGD (Li et al., 2020). Though our design also helps the projected Stiefel SGD to preserve the manifold structure with only machine precision error, the convergence to minimum is slower than both our algorithm and original projected Stiefel SGD.

