# OpenReview forum: "Momentum Stiefel Optimizer, with Applications to Suitably-Orthogonal Attention, and Optimal Transport"
_ICLR.cc/2023/Conference — ICLR 2023 poster_

### Official Review · Reviewer_y3Tv · 2022-10-23

**Confidence:** 1
**Correctness:** 4
**Technical Novelty And Significance:** 3
**Empirical Novelty And Significance:** 3
**Recommendation:** 6

**Clarity, Quality, Novelty And Reproducibility:**

The paper is well structured and well written; The authors provide the code to reproduce the results.

**Strength And Weaknesses:**

The paper is well structured and well written; Since I was not familiar with this field and not able to confirm every single proof/algorithm, I am unable to assess this paper.

**Summary Of The Paper:**

This paper presents a Momentum Stiefel Optimizer with several pleasant properties.

**Summary Of The Review:**

The paper is well structured and well written; Since I was not familiar with this field and not able to confirm every single proof/algorithm, I am unable to assess this paper.

---

### Official Review · Reviewer_TSJE · 2022-10-24

**Confidence:** 4
**Correctness:** 4
**Technical Novelty And Significance:** 2
**Empirical Novelty And Significance:** 4
**Recommendation:** 5

**Clarity, Quality, Novelty And Reproducibility:**

- The paper is very clearly written and easy to read

- The paper is of good quality, the experimental validation is great and convincing. Still, the theoretical analysis is missing.

- The paper proposes an original method to incorporate momentum for optimization on the stiefel manifold.


**Strength And Weaknesses:**

# Strengths
- This paper considers a timely problem : optimization on the stiefel manifold arises naturally in many machine learning problems and deep learning problems, and considering the success of momentum-based techniques  / adam in the Euclidean case, it is important to have extensions of these methods to the stiefel manifold

- the proposed algorithm seems sound, and is computationally quite cheap (on par with classical Riemannian SGD since we have 1 projection per iteration).

- The numerical evaluation is satisfactory, the advantage of the proposed algorithm is clearly illustrated.

# Weakness

- The striking weakness of this paper is the lack of any theoretical result regarding the actual convergence of the proposed method. This is an optimization paper and yet there is no convergence result (while the method seems to work well in practice). This is a clear hole in the plot of the paper: what can we theoretically expect from this method ? Do the iterates of algorithm 1 decrease the loss function?

**Summary Of The Paper:**

This papers considers the problem of optimizing a function on the Stiefel manifold. The main objective of the paper is to build algorithms with momentum, which is often difficult / expensive in a Riemannian context. The key idea of the authors is to consider an ODE with momentum that converges to the solutions of the problem, and then to have a judicious discretization process that produces feasible iterates, while maintaining a relatively low cost (only one projection on the manifold should be computed at each iteration). The authors also extend the celebrated adam algorithm in this setting.

The authors then demonstrate the promises of their approach to train vision transformers with orthogonal attention heads and for computing projection robust wasserstein distances.

**Summary Of The Review:**

This is a paper that this easy and pleasant to read, that proposes a new method for an important problem in ML and that convincingly demonstrates its utility in practice. However, the lack of any theoretical result is problematic to me.

---

### Official Review · Reviewer_RJLh · 2022-10-25

**Confidence:** 3
**Correctness:** 4
**Technical Novelty And Significance:** 3
**Empirical Novelty And Significance:** 3
**Recommendation:** 6

**Clarity, Quality, Novelty And Reproducibility:**

Quality:
Good quality in terms of novel and solidness

Clarity:
Good presentation

Originality:
Novel

**Strength And Weaknesses:**

Strengths:
1) The proposed method looks pretty interesting and novel to me.The authors had a clear presentation from the general idea of constraint optimization on the manifold as a continuous-time process with continuous-time constraints, then to a proper discretization scheme.
2) The problem to be studied is important. The authors have clearly demonstrated that orthogonality helps with the generalization performance of machine learning models, and have demonstrated the benefits of the proposed method with extensive studies.
3) The theoretical analysis and proofs are solid. I only loosely checked the theoretical analysis and it looks pretty convincing to me.

Weakness:
1) It might be better to start with toy examples to introduce the idea of Stiefel optimization, especially for readers like me who are not familiar with this specific field.
2) I would suggest the authors conduct experiments on more challenging datasets rather than Cifar.

**Summary Of The Paper:**

The authors considered the problem of optimization on Stiefel manifold. Heuristically, the proposed method can be divided into the following steps: 1) formulate a variational principle which naturally constraints the trajectory on a Stiefel manifold, hence it exactly preserves the manifold structure without extra operations such as projection / retraction 2) With a smart discretization in time of the derived ODEs, the authors came to an discrete-time optimization scheme.

The provided thorough theoretical analysis, including: 1) detailed derivation of the variational principle 2) proof of the structure preserving property 3) discretization scheme

Finally, the authors studied the benefits of orthogonality, including orthogonality at initialization, within an attention head, and across attention heads. The author validated the proposed method in experiments.

**Summary Of The Review:**

Overall the paper proposed a novel method to solve the Stiefel optimization problem without extra steps like projection / retraction. The idea is novel and well validated.

---

### Official Review · Reviewer_AYdG · 2022-10-25

**Confidence:** 4
**Correctness:** 4
**Technical Novelty And Significance:** 3
**Empirical Novelty And Significance:** 3
**Recommendation:** 10

**Clarity, Quality, Novelty And Reproducibility:**


It is a well written paper. It is naturally heavy on differential geometry of Sitefel manifolds and the formulations of solutions as dynamical systems. Perhaps some of this can be postponed to the appendix.

Also perhaps some early research in computer vision on optimization on Sitefel manifolds can also be cited, especially for comparing with Stochastic Gradient Descent and simulated annealing techniques. See, for example,
Xiuwen Liu, A. Srivastava and K. Gallivan, "Optimal linear representations of images for object recognition," in IEEE Transactions on Pattern Analysis and Machine Intelligence, vol. 26, no. 5, pp. 662-666, May 2004, doi: 10.1109/TPAMI.2004.1273986.


**Strength And Weaknesses:**

Strengths:

This is a strong paper on exploiting the differental geometry of Stiefel manifolds and bringing some of the momentum-based improvements from Euclidean domains to this manifold. The experimental results are quite satisfactory.

**Summary Of The Paper:**

This paper develops a gradient-based solution for optimizations if objective functions on a Stiefel manifold. It uses the geometry of Sitefel manifolds along with momentum-based ideas for speeding up gradient searches to develop an intrinsic solution for the optimization. It is quite elegant in its use of geometry and in ensuring that the iterative updates stay on the required spaces (manifold and the tangent bundle). I did not follow all the details in the construction but trust that the paper is solid in its development. From an application point of view, the focus is on training neural networks where orthogonality is important. The experimental results involve training ViT from scratch using different orthogonality constraints and the proposed method is clearly found to be better.

**Summary Of The Review:**


A strong paper on solving general optimization problems on Stiefel manifolds. Impressive level of details in

---

### Comment · Area_Chair_CatB · 2022-11-18
**Please respond to author rebuttals**

Dear Reviewers,

The authors have submitted their rebuttals. Please have a look and respond to their efforts. This will be a respect to their hard work. Many thanks!

Area Chair

---

> ### Author Response · Authors · 2022-11-18
> **Thank you**
>
> We sincerely thank the area chair, the reviewers, and everyone involved, for working on and helping improve our paper, as well as your precious time.

---

### Comment · Area_Chair_CatB · 2022-11-23
**About Algorithm 3**

Dear Authors,
You wrote above Algorithm 3 that the Newton-Schur iteration presented in Algorithm 3 is quadratically convergent. However, I am sketptic about this as it is written on page 153 of (Higham 1997) that the convergence rate is of order 0, which means that it can only be sublinear at the most. Since Algorithm 3 is critical for reducing the computation cost of your algorithm, could you elaborate on this issue? Thanks!

AC

---

> ### Author Response · Authors · 2022-11-24
> **Convergence speed of Algorithm 3**
>
> Dear Area Chair CatB,
>
> Thank you very much for the question and an opportunity of clarification.
>
> The Higham 1997 paper that we referred to is the one published in Numerical Algorithms 15 (1997) from page 227 to page 242. We were unable to find page 153 where the convergence rate was stated to be of order 0. Higham's eq(2.6b) on page 231 was what we used in Algorithm 3, and right below eq(2.6b) it was written "we have **quadratic** convergence of $Y_k$ to $A^{1/2}$ and $Z_k$ to $A^{-1/2}$". The same statement was repeated several times throughout this reference.
>
> This quadratic convergence was also consistent with our numerical experiments, such as those in Fig.5c.
>
> A side remark is, Algorithm 3 indeed helps us improve the computational efficiency, but it is not the sole reason.  Our speed mainly comes from the automatic structure-preservation of momentum, and quantitative evidence was provided in Appendix C.
>
> Thanks again!
>
> Best wishes,
>
> Paper5483 Authors

---

> > ### Comment · Area_Chair_CatB · 2022-11-25
> > **Thanks for clarification!**
> >
> > Your clarification is fairly enough. What I wrote "Higham 1997" actually referred to his book: Functions of Matrices: Theory and Computation (my fault), where on page 153 it was written that m=0, in which m is the order of convergence stated in Theorem 6.11. Usually, quadratic convergence requires inversion, but there is no inversion in Algorithm 3. That was why I doubted.

---

> > > ### Author Response · Authors · 2022-11-26
> > > **Thank you**
> > >
> > > Thanks for your reply. Indeed, one remarkable fact about Algorithm 3, thanks to Prof. Higham, is its quadratic convergence without even requiring any matrix inversion. The book you mentioned seems to be another very educational reference, and we sincerely appreciate it!

---

### Decision · Program_Chairs · 2023-01-20

**Decision:**

Accept: poster

**Justification For Why Not Higher Score:**

Three reviewers did not give strong support to the paper. They only gave borderline scores.

**Justification For Why Not Lower Score:**

This paper got the highest average scores among the papers I handled. I think the idea of structure preserving discretization and momentum is good.

**Metareview: Summary, Strengths And Weaknesses:**

The paper got one 10 (strong accept), two 6s (marginally above threshold) and one 5 (marginally below threshold). The major weakness of the paper includes that the convergence property was not made clear, missing some references, and the experiments are a bit weak. The authors mostly addressed the reviewers' concerns. By the overall scores and the novelty of the paper, the AC recommended acceptance.

**Note From Pc:**

if the above contains the word "oral" or "spotlight" please see: "oral" presentation means -> notable-top-5% and "spotlight" means -> notable-top-25%. As stated in our emails, we are disassociating presentation type from AC recommendations

**Summary Of Ac-Reviewer Meeting:**

N/A